

# Contrail formation for aircraft with hydrogen combustion - Part 2: Engine-related aspects

Josef Zink[1] and Simon Unterstrasser[1]

[1]Deutsches Zentrum für Luft- und Raumfahrt, Institut für Physik der Atmosphäre, Oberpfaffenhofen, Germany

**Correspondence:** Josef Zink (josef.zink@dlr.de)

**Abstract.** The number of ice crystals formed in nascent contrails strongly influences contrail-cirrus life cycle and radiative forcing. Previous studies on contrails from hydrogen combustion focused on microphysical processes that affect the ice crystal number. These studies, however, paid less attention to engine-related aspects. To fill this gap, we investigate how the exhaust plume evolution is thermodynamically influenced by (i) the engine's overall propulsion efficiency, (ii) the engine exit conditions due to varying ambient conditions, (iii) the engine size and exit jet speed, and (iv) the explicit treatment of kinetic energy dissipation and entrainment of enthalpy initially contained in the bypass flow of a turbofan engine. Based on simulations with the box model version of the Lagrangian Cloud Module, we investigate how these aspects influence the contrail formation process and derive suitable (scaling) relations for the number of ice crystals formed $N_{\text{ice,f}}$ on entrained ambient aerosols for hydrogen combustion. These relations help to derive an expression of $N_{\text{ice,f}}$ through a functional relationship that relies on a reduced set of input parameters, while ensuring a generalized parametrization of $N_{\text{ice,f}}$ in contrails from hydrogen combustion.

## 1 Introduction

### 1.1 Motivation

Aviation's total radiative forcing arises not only from $CO_2$ emissions but also significantly from non-$CO_2$ effects, including nitrogen oxide ($NO_x$) emissions and the formation of persistent contrail cirrus clouds (Lee et al., 2021). The estimation of the contrail cirrus' radiative impact requires a realistic representation of contrails in large-scale models. Therefore, the general circulation model (GCM) ECHAM has been expanded by the contrail module CCMod (Burkhardt and Kärcher, 2009; Bock and Burkhardt, 2016). Due to the coarse resolution of GCMs, small-scale processes in the early stages of contrails can not be explicitly resolved and therefore have to be parametrized. The early ice crystal number is the key quantity of young contrails, which strongly influences the later contrail cirrus properties, life cycle and radiative forcing (Unterstrasser and Gierens, 2010; Lewellen, 2014; Burkhardt et al., 2018; Lottermoser and Unterstrasser, 2025). For kerosene combustion, parametrizations have been developed for both the initial number of ice crystals formed (Kärcher et al., 2015) and the loss of ice crystals due to wake vortex interaction (Unterstrasser, 2016). These parametrizations have been successfully implemented in CCMod





(Bier and Burkhardt, 2019, 2022). To assess the radiative impacts of contrails from alternative aviation technologies such as
hydrogen combustion, these parametrizations need to be revised. Recently, Lottermoser and Unterstrasser (2025) extended the
wake vortex loss parametrization to hydrogen combustion scenarios. However, a parametrization for the initial number of ice
crystals formed is still lacking for hydrogen combustion cases.

## 1.2  Previous research

Contrail formation theories/models range from binary decisions on whether contrails form or not (Schmidt, 1941; Appleman,
1953; Schumann, 1996; Richardson, 2025), to time-squeezed microphysical models (Kärcher et al., 2015), to fully time-
resolved microphysical models. In the latter models, the dynamical representation ranges from 3D large-eddy simulations
(LES) (e.g., Paoli et al., 2008, 2013; Garnier et al., 2014; Vancassel et al., 2014; Lewellen, 2020; Afkari et al., 2025) and
3D Reynolds-Averaged Navier-Stokes (RANS) models (e.g., Khou et al., 2015, 2017; Cantin et al., 2022), to 0D box models
(e.g., Kärcher and Yu, 2009; Bier et al., 2022, 2024; Yu et al., 2024). A 0D box model approach uses an offline approach,
meaning that the microphysics is calculated without feedback on the dynamics. Instead, the dilution, which describes plume
expansion, cooling, and humidity evolution, is externally prescribed. This can be either through analytical formulations (e.g.,
Schumann et al., 1998) or based on output from prior performed LES or RANS simulations. The big advantage of these 0D
box models is their low computational cost due to their simplified dynamic representation, which allows exploring a large
parameter space of emission, aerosol and meteorological parameters with a detailed microphysical contrail formation model.
At their minimal computational cost, when relying on an average trajectory, 0D models neglect any plume heterogeneity.
The plume heterogeneity is accounted for when an ensemble of trajectories (so-called multi-0D approach as described in
Bier et al., 2022) is used, but as long as the box model is run for each trajectory separately, diffusive processes between
nearby trajectories are neglected. Therefore, Lottermoser and Unterstrasser (2024) introduced the 1D framework RadMod as
a compromise between computational expensive 3D models with detailed dynamical core and low-cost 0D box models with
strongly simplified dynamics. So far, only the dynamical component of this 1D framework has been described.

Previous studies on contrail formation from hydrogen combustion showed that microphysical processes strongly control
the number of contrail ice crystals formed (Bier et al., 2024; Zink et al., 2025b, a). For a comprehensive understanding of
contrail formation, however, it is also essential to consider how engine characteristics influence the microphysical processes.
This includes a realistic representation of how various aspects of commercial turbofan engines affect contrail formation, as
well as how these influences may change with advancements in modern and future engine technologies.

Turbofan engines generate thrust through two streams: a core flow, where combustion takes place, and a bypass flow ducted
around the core (Walsh and Fletcher, 2004). The overall propulsion efficiency represents the fraction of the fuel's combustion
heat converted into the work rate performed by the thrust (Schumann, 2000; Schumann et al., 2000). The overall propulsion
efficiency depends on the flight state (Poll and Schumann, 2024) and can be expressed as the product of the thermal efficiency
and the propulsive efficiency. Thermal efficiency measures the proportion of combustion heat converted into power delivered
by the engine core. According to the second law of thermodynamics, this value is always less than one. Thermal efficiency
has increased over the years due to higher overall pressure ratios and higher turbine inlet temperatures (Yin and Rao, 2020).





Current values are around $\sim 0.45$, with a practical upper limit near $\sim 0.55$ due to physical limitations (e.g., material properties) and constraints on $NO_x$ emissions (Yin and Rao, 2020). Propulsive efficiency measures how effectively core delivered power

is converted into thrust. Some energy inevitably remains as kinetic energy in the exhaust, so this efficiency also remains below one. The values of propulsive efficiency have increased over the years, with current values around $\sim 0.8$, and are expected to increase to approximately $\sim 0.85$ (Yin and Rao, 2020). This increase over the years is primarily due to the adoption of higher bypass ratios, which is the ratio of the bypass to core mass flow. A higher bypass ratio means that a larger mass of air is accelerated at a lower velocity reducing the kinetic energy in the exhaust. Bypass ratios have increased from approximately

$\sim 2$ in the 1960s, to $\sim 4 - 6$ in the 1980s, to $\sim 8 - 10$ in the 2000s, and up to around $\sim 12$ in the 2010s with the advent of geared turbofans (Gloeckner and Rodway, 2017; Alves et al., 2020). Future ultra-high bypass ratio engines may reach values up to $\sim 20$ (Gloeckner and Rodway, 2017). However, as the bypass ratio increases, so does the fan diameter, which leads to increased aerodynamic drag and installation challenges (Magrini et al., 2020), imposing a practical upper limit. To overcome this technical limit, a potential future technology could be the use of open rotor fans with bypass ratios $\gtrsim 30$ (Gloeckner and

Rodway, 2017), and theoretical propulsive efficiencies of up to $\sim 0.95$ (Yin and Rao, 2020).

In a comprehensive study, Lewellen (2020) investigated contrail formation for kerosene combustion using a LES and an average 0D box model approach. As part of this study, Lewellen investigated the impact of engine size on dilution speed and developed a corresponding scaling law for the number of ice crystals formed on ambient aerosols. To the best of the authors' knowledge, the influence of engine size has not yet been studied for hydrogen combustion. This aspect could, however, be

of particular relevance, as hydrogen-powered aviation is expected to be introduced first in regional, short- and medium-haul aircraft with smaller engines, and potentially later extended to long-haul aircraft equipped with larger engines (Tiwari et al., 2024; Soleymani et al., 2024).

By comparing a wide range of simulations, Lewellen (2020) also found that a 0D box model approach generally overestimates ice crystal numbers relative to LES results when simulating contrail formation on emitted particles with high numbers.

In contrast, box model results closely match the LES outcomes for low particle numbers or contrail formation on entrained ambient aerosols (which is the case in our setups). Hence, a 0D box model approach is deemed appropriate for our study, which deals with contrail formation on entrained ambient aerosols for hydrogen combustion.

### 1.3 Scope and outline of the study

Here we literally repeat the according subsection "Scope and outline of the study" of Part 1 (see Sect. 1.5 in Zink et al. (2025a)):

"Our overarching goal is to develop a parameterization for the final number of ice crystals formed for aircraft with hydrogen combustion, suitable for implementation in GCMs and other large-scale contrail models. Specifically, we seek a functional relationship of the form

$$N_{\text{ice,f}} = N_{\text{ice,f}}(\mathbf{a}_{\text{atmosphere}}, \mathbf{a}_{\text{aerosol}}, \mathbf{a}_{\text{aircraft}}) , \tag{1}$$





where $\mathbf{a}_{\mathrm{atmosphere}}$ denotes a set of parameters characterizing the background atmosphere (e.g., ambient temperature), $\mathbf{a}_{\mathrm{aerosol}}$
represents properties of ambient aerosols (number concentration, size distribution and solubility) and $\mathbf{a}_{\mathrm{aircraft}}$ includes aircraft-related parameters (e.g., engine size).

Our objective is to identify a functional relationship that balances simplicity with physical fidelity. In other words, we aim to capture the impact of the key processes that govern ice crystal formation in a form that is simple enough to be easily implemented in other models. In Parts 1 and 2 of a trilogy of papers, we systematically explore a broad parameter space and address aspects that have not been explored before to gain a deep insight into the physical processes influencing the ice crystal formation. This allows us to select the set of parameters that constitute the inputs to the $N_{\mathrm{ice,f}}$ parametrization.

While a previous $N_{\mathrm{ice,f}}$ parametrization of contrail formation (Kärcher et al., 2015) used a first-principles-based concept, the current work across Parts 1 to 3 employs a hybrid approach. It combines a data-driven advanced regression method to fit a multidimensional database of contrail formation simulations with analytical scaling relations. To keep the number of dimensions in the simulation database as small as possible, we identify conditions under which the sensitivity to specific parameters is negligible, and also employ analytical scaling relations derived from sensitivity simulations.

In Part 1, we focus on the roles of atmospheric parameters $\mathbf{a}_{\mathrm{atmosphere}}$ and aerosol parameters $\mathbf{a}_{\mathrm{aerosol}}$. Aircraft-related parameters $\mathbf{a}_{\mathrm{aircraft}}$ are addressed in Part 2. The final parameterization, synthesizing insights from Parts 1 and 2, will be presented in Part 3."

In the current study, we incorporate the leading effects of engine-related aspects into the simulations. In Sec. 2 we provide the theoretical background about the extensions made to our used box model (described in Sec. 3). We then investigate the impact of these aspects on the number of ice crystals formed (Sec. 4) before we discuss the findings in the light of existing literature and propose possibilities for future research (Sec. 5). Finally, we conclude with a discussion of how our findings can be used in the development of a generalized parametrization of ice crystal number (Sec. 6).

## 2   Theoretical considerations

In this section, we describe how various engine-related aspects influence the thermodynamic evolution of the exhaust plume. These considerations are not restricted to the specific model used. Nevertheless, they motivate and inform the extensions made to our box model (described in Sec. 3). Furthermore, the considerations are valid for both hydrogen and kerosene combustion and do not involve microphysics, making them, for example, independent of assumptions about the type of ice-forming particles.

In a first step, we shortly summarize the most relevant equations describing the classical thermodynamic contrail formation theory (Sec. 2.1). We then present theoretical estimates of how changes in overall propulsion efficiency (Sec. 2.2), ambient temperature and pressure as well as jet velocity and size (Sec. 2.3) influence the jet dilution and plume thermodynamics. Finally, we introduce a framework that allows to consider the effect of the initial separation of the exhaust into a core and bypass flow on the plume thermodynamical evolution in our box model (Sec. 2.4).



## 2.1 Classical thermodynamic theory

We review and present the equations necessary for understanding the considerations in the subsequent sections. Detailed derivations and explanations can be found in Bier et al. (2022, 2024).

Fuel is combusted in an aircraft engine to produce thrust, propelling the aircraft forward in accordance with Newton's third law of motion. In this context, the overall propulsion efficiency (Schumann, 2000)

$$\eta = \frac{FU_\infty}{\dot{m}_{\mathrm{F}}Q} \tag{2}$$

is defined as the ratio of work rate performed by the thrust $F$ at aircraft speed $U_\infty$ and the chemical energy supplied by the fuel, which is determined by the fuel flow rate $\dot{m}_{\mathrm{F}}$ and combustion heat $Q$ (also known as lower calorific value). The remaining part of the chemical energy $\dot{m}_{\mathrm{F}}Q(1-\eta)$, which does not contribute to generating the thrust, is contained in the exhaust air in the form of excess total enthalpy above ambient levels. The total enthalpy is the sum of static enthalpy and kinetic energy. Additionally, the combustion of fuel adds water vapor to the core exhaust, quantified by the fuel-dependent water vapor (mass) emission index $EI_{\mathrm{v}}$.

We refer to the core exhaust as plume and to individual parts of it as plume parcels. Each plume parcel mixes with cold ambient air downstream of the engine. Classical thermodynamic contrail formation theory assumes that the core and bypass air of a turbofan engine are fully mixed by the time contrail formation begins (or, equivalently, that the emitted enthalpy is entirely contained within the core flow at the engine exit). Additionally, it assumes a stagnant exhaust plume, such that the entire kinetic energy is presumed to have been dissipated into heat at the onset of contrail formation. Furthermore, the theory assumes the same entrainment rate $\omega$ for both enthalpy and mass, implying a Lewis number of one. This assumption is generally reasonable, as turbulent diffusion is expected to dominate over molecular diffusion under contrail formation conditions within a strong jet. The entrainment rate is defined as

$$\omega(t) = -\frac{\mathrm{d}\ln\mathcal{D}}{\mathrm{d}t} \ , \tag{3}$$

where $\mathcal{D}$ is the dilution factor. The value of $\mathcal{D}$ is one at the engine exit and gives the mass fraction of core air in a plume parcel downstream of the engine (hence, it is a monotonically decreasing function with time/downstream position).

In addition to the stated assumptions, an isobaric mixing process is assumed. Furthermore, when relating static enthalpy and temperature, the slight temperature dependence of the specific heat capacity at constant pressure $c_{\mathrm{p}}$ is neglected (Schumann, 1996). The temperature evolution of a plume parcel is then governed by (Kärcher, 1995)

$$\frac{\mathrm{d}T}{\mathrm{d}t} = -\omega(t)\left(T(t) - T_{\mathrm{a}}\right) + S_{\mathrm{LH}}(t) \ . \tag{4}$$

In Eq. (4), $T_{\mathrm{a}}$ is the ambient temperature and $S_{\mathrm{LH}}$ denotes a source/sink term accounting for the microphysical latent heat release/consumption during phase transitions. Analogously, the equation for the plume parcel's water vapor mass fraction $\chi_{\mathrm{v}}$ is expressed as

$$\frac{\mathrm{d}\chi_{\mathrm{v}}}{\mathrm{d}t} = -\omega(t)\left(\chi_{\mathrm{v}}(t) - \chi_{\mathrm{v,a}}\right) + S_{\mathrm{PH}}(t) \ , \tag{5}$$



where $\chi_{v,a}$ is the ambient level and $S_{PH}$ is a source/sink term due to phase changes. The plume parcel's water vapor mass fraction $\chi_v$ and partial water vapor pressure $e_v$ are related by (Schumann, 1996)

$$e_v = \frac{p_a}{\epsilon}\chi_v \tag{6}$$

with the ambient pressure $p_a$ and the ratio of molar masses of dry air and water vapor $\epsilon = 0.622$.

Eqs. (4) to (6) lead for each plume parcel in the absence of microphysical processes ($S_{LH} = S_{PH} = 0$) to an universal mixing line in an $e_v$-$T$ diagram. The slope of this mixing line is determined by (Schumann, 1996)

$$G = \frac{EI_v c_p p_a}{\epsilon Q(1-\eta)} \ . \tag{7}$$

The entrainment rate $\omega$ just tells how fast the thermodynamic quantities describing the plume parcel's state evolve along this
line.

At engine exit, the exhaust is already a mixture of ambient air and fuel mass, which is described by the air-to-fuel ratio

$$\mathcal{C}_E = \frac{Q(1-\eta)}{c_p(T_E - T_a)} \ , \tag{8}$$

with $T_E$ denoting the exit temperature. The time-dependent overall dilution of a plume parcel is then defined as

$$\begin{aligned}\mathcal{C}(t) &= \frac{\mathcal{C}_E + 1}{\mathcal{D}(t)} \\ &\approx \frac{\mathcal{C}_E}{\mathcal{D}(t)} \ , \end{aligned} \tag{9}$$

where the fuel mass is neglected in the approximation.

The fuel mass flow rate can be expressed as

$$\dot{m}_F = \frac{p_a \hat{A}_E U_\infty}{R_d T_E \mathcal{C}_E} \tag{10}$$

with the specific gas constant $R_d$ and effective core exit area

$$\hat{A}_E = A_E \frac{U_{tot,E}}{U_\infty} \ . \tag{11}$$

The effective core exit area $\hat{A}_E$ combines the true physical exit area $A_E$ with the ratio between total exit jet speed $U_{tot,E}$ and aircraft speed $U_\infty$. The total jet speed $U_{tot,E}$ is the sum of excess jet speed $U_{jet,E}$ with respect to the ambient air and the aircraft speed $U_\infty$. The effective area accounts for the axial divergence of the jet and is representative of a plume parcel's volume (see Sec. 3.2.2 in Bier et al. (2024) for details). Using $\hat{A}_E$, the number of entrained ambient aerosols per meter of flightpath is given by

$$N_{aer}(t) = n_{aer} \frac{T_a \hat{A}_E}{T_E} \left(\mathcal{D}^{-1}(t) - 1\right) \ , \tag{12}$$

where $n_{aer}$ is the ambient aerosol number concentration.





## 2.2 Variation of overall propulsion efficiency

We aim to study the impact of varying supersaturations that arise from the influence of $\eta$ on the slope $G$ of the mixing line (Eq. (7)). Clearly, an increase in $\eta$ reflects technological advancements in engine configuration (Sec. 1.2). Here, we do not explicitly address the underlying causes of varying efficiencies among different engines (e.g., differences in engine design such as bypass ratios or jet velocities), and we therefore neglect their potential influence on exhaust plume dynamics and the associated dilution speed. Such a dilution time scaling is proposed in Sec. 2.3 for other purposes. Instead, our focus is to investigate the thermodynamic influence of $\eta$ on the formation of ice crystals. To this end, we present a physically consistent model setup as follows.

We assume the same ambient conditions and the same flight Mach number but an engine that is more/less efficient, i.e., a change of overall propulsion efficiency from a reference value $\eta^*$ to another value $\eta$. According to Eq. (2) this implies that the fuel flow has to be adjusted to

$$\dot{m}_{\mathrm{F}} = \frac{\eta^*}{\eta} \dot{m}_{\mathrm{F}}^* \tag{13}$$

to maintain the thrust. We now assume that the core mass flow is preserved, as then the plume's mass and number of entrained ambient particles (Eq. (12)) stay the same for a given dilution state downstream of the engine. From preserving the core mass flow, a change in the air-to-fuel ratio

$$\mathcal{C}_{\mathrm{E}} = \frac{\eta}{\eta^*} \mathcal{C}_{\mathrm{E}}^* \tag{14}$$

follows. In combination with Eq. (8) this implies a change in core engine exit temperature

$$T_{\mathrm{E}} = \frac{\eta^*(1-\eta)}{\eta(1-\eta^*)} \left( T_{\mathrm{E}}^* - T_{\mathrm{a}} \right) + T_{\mathrm{a}} \; , \tag{15}$$

where $T_{\mathrm{a}}$ is the ambient temperature. Eqs. (13) to (15) together with Eq. (10) imply a change in the core exit effective area

$$\hat{A}_{\mathrm{E}} = \frac{T_{\mathrm{E}}}{T_{\mathrm{E}}^*} \hat{A}_{\mathrm{E}}^* \; . \tag{16}$$

### 2.3 Dilution time scaling

The length and velocity scale of a free turbulent round jet are the nozzle exit radius $r_{\mathrm{E}}$ and exit velocity $U_{\mathrm{jet,E}}$, respectively (Pope, 2000). We write down the Reynolds-averaged mean momentum equation in polar-coordinates (Kärcher and Fabian, 1994; Lottermoser and Unterstrasser, 2024), which reads in non-dimensional form

$$\hat{\rho}\hat{U}\frac{\partial \hat{U}}{\partial \hat{x}} + \hat{\rho}\hat{V}\frac{\partial \hat{U}}{\partial \hat{r}} = \frac{\hat{D}_{\mathrm{T}}}{\hat{r}}\frac{\partial}{\partial \hat{r}}\left( \hat{\rho}\hat{r}\frac{\partial \hat{U}}{\partial \hat{r}} \right) \tag{17}$$

with the scaled axial velocity $\hat{U} = U/U_{\mathrm{jet,E}}$, radial velocity $\hat{V} = V/U_{\mathrm{jet,E}}$, axial coordinate $\hat{x} = x/r_{\mathrm{E}}$, radial coordinate $\hat{r} = r/r_{\mathrm{E}}$, density $\hat{\rho} = \rho/\rho_0$, and turbulent viscosity $\hat{D}_{\mathrm{T}} = D_{\mathrm{T}}/(r_{\mathrm{E}}U_{\mathrm{jet,E}})$.



Since $\hat{D}_{\mathrm{T}} \approx \mathrm{const.}$ (Pope, 2000), solving Eq. (17) together with the continuity equation, temperature equation, and equation of state gives a universal solution for the scaled variables. Looking at a time increment

$$\mathrm{d}t = \frac{\mathrm{d}x}{U} = \frac{\mathrm{d}\hat{x}}{\hat{U}} \frac{r_{\mathrm{E}}}{U_{\mathrm{jet,E}}} \tag{18}$$

and setting $\mathrm{d}\hat{x} = \mathrm{d}\hat{x}^* \cdot 1$ and $\hat{U} = \hat{U}^* \cdot 1$ (due to the universal solution for the scaled variables), it follows the relation

$$\begin{aligned}
\mathrm{d}t &= \frac{U_{\mathrm{jet,E}}^*}{U_{\mathrm{jet,E}}} \cdot \frac{r_{\mathrm{E}}}{r_{\mathrm{E}}^*} \cdot \mathrm{d}t^* \\
&= s_{\mathrm{shear}} \cdot s_{\mathrm{E}} \cdot \mathrm{d}t^* \\
&= s_{\mathrm{dil}} \cdot \mathrm{d}t^* \,,
\end{aligned} \tag{19}$$

where asterisks symbols denote reference values. In Eq. (19) we use the definitions

$$s_{\mathrm{shear}} := U_{\mathrm{jet,E}}^*/U_{\mathrm{jet,E}} \tag{20a}$$
$$s_{\mathrm{E}} := r_{\mathrm{E}}/r_{\mathrm{E}}^* \tag{20b}$$
$$s_{\mathrm{dil}} := s_{\mathrm{shear}} \cdot s_{\mathrm{E}} \,. \tag{20c}$$

This dilution time scaling (Eq. (19)) was first proposed by Lewellen (2020). By using $U_{\mathrm{jet,E}}$ as velocity scale, we implicitly assume a strong jet speed regime (Chu et al., 1999), where the jet dynamics are similar to a setup without relative movement between jet source (here aircraft engine) and environment. Far downstream of the aircraft engine, however, when the jet has been decelerated, the dilution is influenced by the coflow velocity $U_\infty$ (Chu et al., 1999; Kärcher and Fabian, 1994; Enjalbert et al., 2009; Lottermoser and Unterstrasser, 2024). Moreover, $\hat{D}_{\mathrm{T}} \approx \mathrm{const.}$ might be only a valid assumption when a self-similar flow has been established (Pope, 2000; Lottermoser and Unterstrasser, 2024). In addition, the dynamical influence of a bypass flow is not explicitly treated in the above derivation. Nevertheless, as suggested by Lewellen (2020), Eq. (19) may contain the leading factors of a dilution time scaling. In the analysis of our simulation results, the scaling properties presented so far will help to reveal underlying and possibly hidden relations between seemingly different scenarios.

### 2.3.1 Impact of ambient conditions on engine exit conditions

In many cases, we set up box model simulations using ambient conditions (ambient temperature $T_{\mathrm{a}}$ and ambient pressure $p_{\mathrm{a}}$) that differ from those used in the Computational Fluid Dynamics (CFD) simulations ($T_{\mathrm{a}}^*$ and $p_{\mathrm{a}}^*$) from which the dilution factor is extracted. The ambient conditions, in turn, influence the engine exit conditions and jet dilution speed. To account for this, we follow the procedure recommended by Lewellen (2020). He suggests preserving temperature and pressure ratios, Mach numbers, and overall propulsion efficiency following common industry practice (Volponi, 1998; Kurzke, 2003; Walsh and Fletcher, 2004). This leads to the scaling relations summarized in Table 1. Note that the scaling refers to total temperatures and pressure values (defined in Appendix A). However, neglecting the slight dependency of the adiabatic index $\kappa$ on the temperature, this holds also for static temperature and pressure values.





We use the scaling relations in Table 1 to prescribe the engine exit conditions in the box model (e.g., $T_\mathrm{E} = T_\mathrm{a}/T_\mathrm{a}^* \cdot T_\mathrm{E}^*$). The velocity scale follows from preserving Mach numbers together with the fact that the speed of sound scales with $T^{1/2}$. According to this velocity scale $(T_\mathrm{a}/T_\mathrm{a}^*)^{1/2}$ and Eq. (19), the dilution time scales as

$$s_\mathrm{dil} = s_\mathrm{shear} = \left(\frac{T_\mathrm{a}^*}{T_\mathrm{a}}\right)^{1/2} . \tag{21}$$

**Table 1.** Scales for varying ambient conditions. The subscript 'a' stands for ambient. Asterisk denote reference values at which the CFD simulation was performed.

| **variable** | temperatures | pressures | densities | velocities | fuel flow |
|---|---|---|---|---|---|
| **scale** | $T_\mathrm{a}/T_\mathrm{a}^*$ | $p_\mathrm{a}/p_\mathrm{a}^*$ | $(p_\mathrm{a}/p_\mathrm{a}^*) \cdot (T_\mathrm{a}^*/T_\mathrm{a})$ | $(T_\mathrm{a}/T_\mathrm{a}^*)^{1/2}$ | $p_\mathrm{a}/p_\mathrm{a}^* \cdot (T_\mathrm{a}/T_\mathrm{a}^*)^{1/2}$ |

### 2.3.2 Engine size scaling

Assuming an otherwise comparable engine (unchanged temperatures, pressures, velocities, bypass ratio, overall propulsion efficiency, and cruise conditions), but scaling the exit radius $r_\mathrm{E}$ by a factor $s_\mathrm{E}$ leads to the following scaling relations (Lewellen, 2020): The exit area scales with $s_\mathrm{E}^2$, and accordingly also the fuel flow and thrust if everything else stays the same. Moreover, the dilution time scales with $s_\mathrm{dil} = s_\mathrm{E}$ (Eq. (19)). Although $s_\mathrm{dil}$ has the same value as $s_\mathrm{E}$ in this consideration, we explicitly distinguish between them to disentangle the geometric effect from the effect on the dilution.

Clearly, scaling the same engine in size is a rather academically-driven approach and physical and technical limitations have to be considered in real-world engine design (see Lewellen (2020) for details). Nevertheless, the leading impact of engine size on the number of ice crystals formed can be studied with this approach.

### 2.4 Treatment of bypass flow and kinetic energy dissipation

As the combustion takes place in the core of a turbofan engine, excess water vapor at the engine exit is only contained in the core flow. However, part of the chemical energy released during combustion is used to drive the fans. Due to the work performed on the fluid by the blades, the bypass air contains excess static enthalpy and kinetic energy above ambient levels. Consequently, water vapor and total enthalpy are distributed differently at the engine exit of a turbofan engine.

Richardson (2025) investigated contrail formation criterion for various propulsion systems. Assuming that the core and bypass of a turbofan are already well-mixed at engine exit, this mixed exhaust is already supersaturated in his investigated hydrogen combustion case (see his Fig. 4b). As this well-mixed state is one of the assumptions of the classical thermodynamic contrail formation theory (Sec. 2.1), this approach becomes insufficient when simulating the time-resolved contrail formation process. Therefore, a more sophisticated modeling approach is needed accounting for the inital separation of emitted total enthalpy into the core and bypass flow.





### 2.4.1 Revised temperature equation

The goal of this section is to derive a revised temperature equation when two main assumptions of the classical thermodynamic theory (Sec. 2.1) are dropped: The first assumption is that all kinetic energy has been completely converted into heat at the onset of contrail formation and the second is that the core and bypass air of a turbofan engine are already fully mixed at this stage.

To describe the initial separation of emitted total enthalpy into the core and bypass flow, we introduce the (time-constant) scalar factor

$$
\begin{aligned}
\beta &= \frac{E_{\text{tot,core}}\dot{m}_{\text{core}}}{E_{\text{tot,core}}\dot{m}_{\text{core}} + E_{\text{tot,bypass}}\dot{m}_{\text{bypass}}} \\
&= \frac{E_{\text{tot,core}}}{E_{\text{tot,core}} + E_{\text{tot,bypass}} \cdot b}
\end{aligned}
\tag{22}
$$

with the core and bypass mass flows $\dot{m}_{\text{core}}, \dot{m}_{\text{bypass}}$, the bypass ratio $b = \dot{m}_{\text{bypass}}/\dot{m}_{\text{core}}$ and the mass-specific total enthalpies at engine exit

$$
\begin{aligned}
E_{\text{tot},y} &= E_{\text{static},y} + E_{\text{kin},y} \\
&= c_{\text{p}}(T_{\text{E},y} - T_{\text{a}}) + \frac{1}{2}\left(U_{\text{tot,E},y} - U_{\infty}\right)^2 \ ,
\end{aligned}
\tag{23}
$$

which are evaluated in a frame of reference fixed to the ambient air (Schumann, 2000). In Eq. (23), $y$ denotes core or bypass, $c_{\text{p}}$ is the specific heat capacity at constant pressure, $T_{\text{E},y}$ denotes exit temperature, $T_{\text{a}}$ is the ambient temperature and $U_{\text{tot,E},y}$ is the sum of excess jet velocity $U_{\text{jet,E},y}$ and freestream velocity $U_{\infty}$. Note that the calculation of the static enthalpy with $c_{\text{p}}(T_{\text{E},y} - T_{\text{a}})$ is a simplification of the formula $\int_{T_{\text{a}}}^{T_{\text{E},y}} c_{\text{p}}(\tilde{T})\mathrm{d}\tilde{T}$ neglecting the temperature-dependency of $c_{\text{p}}$ (Schumann, 1996) and assumes that the static pressure at engine exit has already relaxed to the ambient pressure.

The factor $\beta$ can take values between zero and one. The term $\beta\dot{m}_{\text{F}}Q(1-\eta)$ is the part of emitted combustion heat that is contained in the core flow at the engine exit. The remaining part $(1-\beta)\dot{m}_{\text{F}}Q(1-\eta)$ is contained in the bypass flow distributed over a $b$ times larger air mass.

Therefore, downstream of the engine, three air masses with initially different properties mix: Hot core air containing excess water vapor, moderately heated bypass air without a water vapor add-on, and cold ambient air with the ambient moisture level. Each air parcel downstream of the engine contains a specific fraction of each of the three air masses. Since excess water vapor (the prerequisite for contrails) is only contained in the core flow, we define an air parcel that contains core air as a 'plume parcel'. This is in accordance with the definition of the dilution factor $\mathcal{D}$ as mass fraction of core air.

We define the time-dependent factor

$$
\gamma(t) := \frac{1}{b}\frac{\mathcal{B}(t)}{\mathcal{D}(t)}
\tag{24}
$$

as the ratio between the fraction of bypass air in the plume parcel $\mathcal{B}$, and the fraction of core air in the plume parcel $\mathcal{D}$ normalized by the bypass ratio $b$. It holds $\gamma = 0$ at core engine exit and $\gamma \to 1$, when core and bypass air are well mixed. With





the definition of $\gamma$, we can write down the equation for the plume parcel's mass-specific total enthalpy for an isobaric mixing process as

$$E_{\text{tot}}(t) := c_{\text{p}}\left(T(t) - T_{\text{a}}\right) + E_{\text{kin}}(t) = \frac{Q(1-\eta)\cdot[\beta+(1-\beta)\gamma(t)] - h_{\text{a}}}{\mathcal{C}(t)}$$

$$\approx \frac{Q(1-\eta)\cdot[\beta+(1-\beta)\gamma(t)]}{\mathcal{C}(t)} \tag{25}$$

with the plume parcel's static temperature $T$, kinetic energy $E_{\text{kin}}$ and the overall dilution $\mathcal{C}$ (Eq. (9)). Here, we neglect the ambient static enthalpy $h_{\text{a}}$ on the right-hand side of Eq. (25) (as done in Schumann (1996)). Furthermore, we just write $E_{\text{kin}}$ instead of the explicit form as done in Eq. (23) for reasons explained in Sec. 2.4.2. According to Eq. (25), the plume parcel's mass-specific total enthalpy is determined by the diluted mass-specific core enthalpy $\beta Q(1-\eta)/\mathcal{C}$ and by the diluted mass-specific bypass enthalpy $(1-\beta)Q(1-\eta)\gamma/\mathcal{C}$, which has been gradually mixed into the plume parcel.

Evaluation of Eq. (25) at core engine exit (denoted by the subscript 'E', the subscript 'core' is omitted in the following) gives the air-to-fuel ratio

$$\mathcal{C}_{\text{E}} = \frac{\beta Q(1-\eta)}{c_{\text{p}}(T_{\text{E}} - T_{\text{a}}) + E_{\text{kin,E}}} \ . \tag{26}$$

Note that compared to Eq. (8), the factor $\beta$ and $E_{\text{kin,E}}$ arise additionally in Eq. (26).

Isolating the term $Q(1-\eta)$ in Eqs. (25) and (26) and equating both equations, leads to the generalized state equation of plume temperature

$$T(t) = T_{\text{a}} - \frac{E_{\text{kin}}(t)}{c_{\text{p}}} + \mathcal{D}(t)\cdot\left[(T_{\text{E}} - T_{\text{a}}) + \frac{E_{\text{kin,E}}}{c_{\text{p}}}\right]\cdot\frac{\beta+(1-\beta)\gamma(t)}{\beta} \tag{27}$$

at any time instance. Taking the logarithmic derivative in time of Eq. (27) leads together with the definition of the entrainment rate (Eq. (3)) to the general ODE for plume temperature

$$\frac{\text{d}T}{\text{d}t} = \left[-\omega(t) + \frac{(1-\beta)\frac{\text{d}\gamma}{\text{d}t}}{\beta+(1-\beta)\gamma(t)}\right]\cdot\left[T(t) - T_{\text{a}} + \frac{E_{\text{kin}}(t)}{c_{\text{p}}}\right] - \frac{1}{c_{\text{p}}}\frac{\text{d}E_{\text{kin}}}{\text{d}t} + S_{\text{LH}}(t)$$

$$= -\hat{\omega}(t)\left[T(t) - T_{\text{a}} + \frac{E_{\text{kin}}(t)}{c_{\text{p}}}\right] - \frac{1}{c_{\text{p}}}\frac{\text{d}E_{\text{kin}}}{\text{d}t} + S_{\text{LH}}(t) \tag{28}$$

with $\hat{\omega} = \omega - (1-\beta)\,\text{d}\gamma/\text{d}t\,[\beta+(1-\beta)\gamma]^{-1}$. In Eq. (28), we added a source/sink term $S_{\text{LH}}$ accounting for the microphysical latent heat release/consumption during phase transitions.

Compared to Eq. (4), Eq. (28) includes two additional physical processes. Firstly, the adapted entrainment rate $\hat{\omega}$ includes the effect of the gradual entrainment of the enthalpy initially contained in the bypass. Secondly, it accounts for the entrainment of bypass kinetic energy into the plume and for the viscous heating within the plume via the derivative of kinetic energy. As soon as core and bypass air are well mixed ($\gamma \to 1$, $\text{d}\gamma/\text{d}t \to 0$) and the kinetic energy has dissipated into heat ($E_{\text{kin}} \to 0$, $\text{d}E_{\text{kin}}/\text{d}t \to 0$), Eq. (28) reduces to Eq. (4). Since water vapor is released only in the core flow, the differential equation for the water vapor mass fraction $\chi_{\text{v}}$ is fully described by Eq. (5) without adaptations needed.

As stated in Sec. 2.1, the classical thermodynamic theory leads to an universal mixing line for all plume parcels. In contrast, when considering explicitly the mixing of core and bypass air (Eq. (28)), each plume parcel can follow a distinct mixing curve





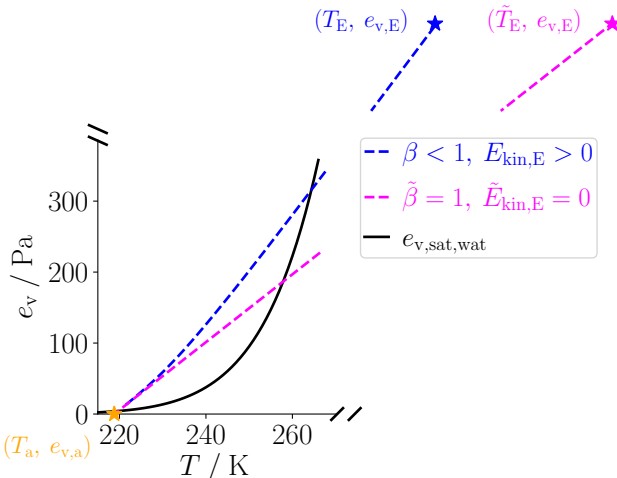

**Figure 1.** Exemplary mixing curve (plume partial pressure of water vapor $e_v$ vs temperature $T$) with an explicit treatment of kinetic and bypass energy (blue) compared to the universal mixing line for a stagnant plume (magenta), where the emitted combustion heat is assumed to be contained in form of static enthalpy in the core flow at engine exit. Stars denote exit and ambient conditions, respectively. The exit temperature for the stagnant plume has to be adjusted after Eq. (29) to ensure the same energy input into the system. The black curve depicts the saturation water vapor pressure over water. While this figure is intended to serve as an illustrative example, it is based on the values taken from the RANS data presented in Sec. 2.4.2.

determined by the time evolution of $\mathcal{D}$, $\gamma$ and $E_{\mathrm{kin}}$. However, each mixing curve converges towards the universal mixing line when core and bypass air are well-mixed and kinetic energy has dissipated into heat downstream of the engine.

Therefore, we intend to compare simulations incorporating the general ODE for plume temperature (Eq. (28)) to the more simplified approach, where we assume that all emitted combustion heat is already contained in form of static enthalpy in the
315 core at engine exit, i.e., a stagnant plume with $\tilde{\beta} = 1$ and $\tilde{E}_{\mathrm{kin,E}} = 0$. If we kept the same core exit temperature, this would imply a higher air-to fuel ratio (Eq. (26) with $\tilde{\beta} = 1$ and $\tilde{E}_{\mathrm{kin,E}} = 0$) and would wholly neglect the kinetic energy and the enthalpy contained in the bypass flow at engine exit. Therefore, a physically more reasonable approach is to prescribe the same air-to fuel ratio $\tilde{\mathcal{C}}_{\mathrm{E}} = \mathcal{C}_{\mathrm{E}}$, which means a change in core exit temperature

$$\tilde{T}_{\mathrm{E}} = T_{\mathrm{a}} + \frac{T_{\mathrm{E}} - T_{\mathrm{a}}}{\beta} + \frac{E_{\mathrm{kin,E}}}{c_{\mathrm{p}}\beta} \quad . \tag{29}$$

In this approach, the total enthalpy input into the system is the same as for the case with explicit treatment of the bypass flow and kinetic energy. The effective core exit area, however, has to be adjusted according to Eq. (16) to have the same core air mass flow into the system, such that the number of entrained aerosols is the same in both approaches (Eq. (12)). These two approaches are illustrated in Fig. 1 for an average trajectory (see Sec. 2.4.2 for a proper definition).





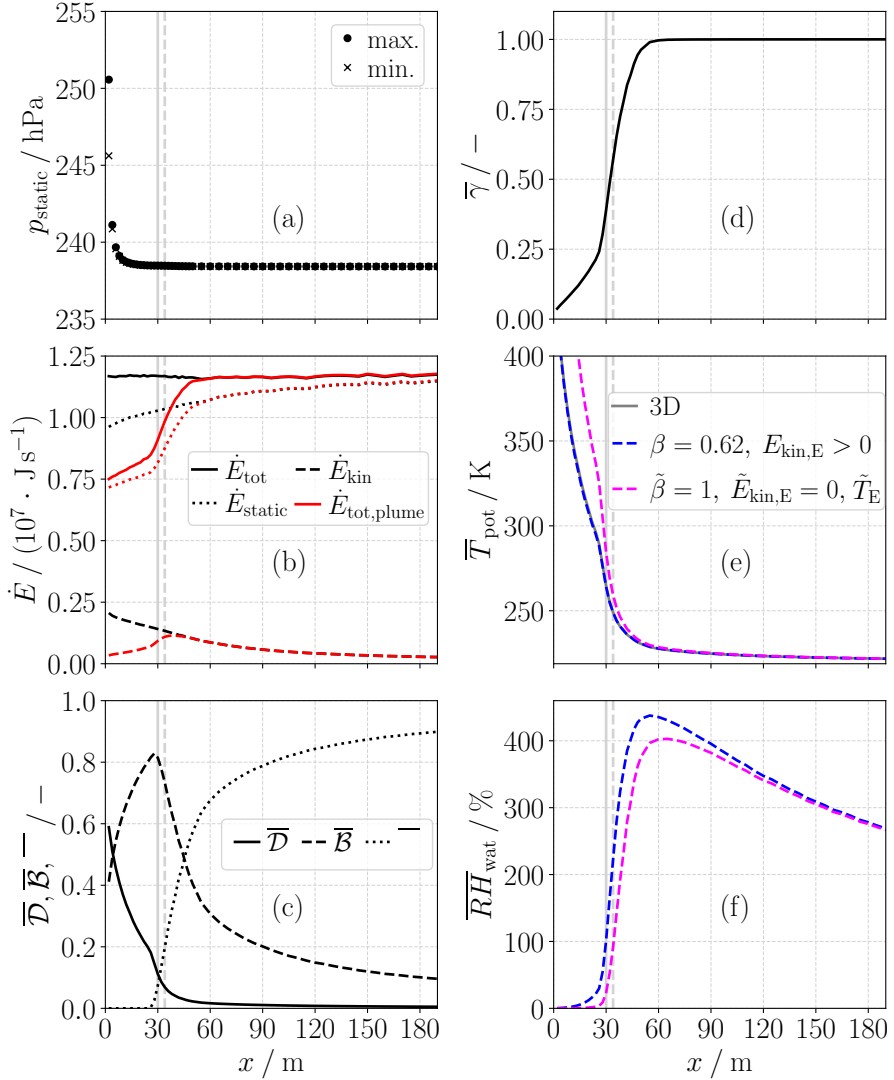

**Figure 2.** Evaluation of RANS data for the Airbus Turbofan engine as a function of downstream distance $x$ with boundary conditions as listed in Table 2. (a) Maximum and minimum static pressure $p_{\text{static}}$. (b) Total enthalpy flow $\dot{E}_{\text{tot}}$ as sum of static enthalpy flow $\dot{E}_{\text{static}}$ and kinetic energy flow $\dot{E}_{\text{kin}}$ evaluated over the whole exhaust air, i.e., bypass and core air (black). The red curves show the total enthalpy flow and its partition evaluated within the plume defined via the core air. (c) Mean fractions of core air (dilution factor) $\overline{\mathcal{D}}$, bypass air $\overline{\mathcal{B}}$ and ambient air $\overline{\mathcal{A}}$ within the plume. (d) Mean $\overline{\gamma}$ factor defined in Eq. (24). (e) Mean plume (potential) temperature $\overline{T}_{\text{pot}}$ extracted from the 3D field (grey curve) and calculated with the generalized state equation of plume temperature Eq. (27) (blue dashed curve) inserting mean quantities $\overline{E}_{\text{kin}}$, $\overline{\mathcal{D}}$ and $\overline{\gamma}$. The magenta curve shows the temperature evolution where the emitted combustion heat is assumed to be contained as static enthalpy in the core flow at engine exit with adjusted exit temperature after Eq. (29). (f) Mean plume relative humidity over water $\overline{RH}_{\text{wat}}$ for the latter two scenarios in (e) calculated with mean quantities assuming a dry environment. For the two scenarios, the grey vertical lines indicate the onset of plume supersaturation $\overline{RH}_{\text{wat}} > 100\%$ and subsequent contrail formation (the same lines are depicted in each panel).



### 2.4.2 Evaluation with RANS data

Airbus provided us with RANS simulation data for an isolated generic turbofan engine burning hydrogen and flying at an ambient temperature $T_{\mathrm{a}} = 218.8\,\mathrm{K}$, ambient pressure $p_{\mathrm{a}} = 238.42\,\mathrm{hPa}$ with an aircraft speed $U_{\infty} = 231\,\mathrm{m\,s^{-1}}$. For the simulation they used the compressible multi-species finite volume solver FLUSEPA, developed by the ArianeGroup (Brenner, 1991; Pont et al., 2017). Passive tracers have been seeded into the core and bypass flow to track the fractions of the different air masses (core, bypass, ambient) within each grid box at any downstream position. We obtained a 3D volume of data covering both core and bypass flow, starting just downstream of the engine exit and ranging up to a downstream distance of about $300\,\mathrm{m}$.

Pressure levels above the background are encountered near the engine exit (Fig. 2a). Since these fluctuations are relaxed to the background when contrail formation starts, the assumption of an isobaric mixing process during contrail formation is justified. Nevertheless, in order to evaluate enthalpy flows near the engine exit, we calculate potential temperature, 'potential' density, and 'potential' velocity as

$$T_{\mathrm{pot}} = T \left( \frac{p_{\mathrm{a}}}{p} \right)^{R_{\mathrm{d}}/c_{\mathrm{p}}} \tag{30a}$$

$$\rho_{\mathrm{pot}} = \frac{p_{\mathrm{a}}}{R_{\mathrm{d}} T_{\mathrm{pot}}} \tag{30b}$$

$$U_{\mathrm{tot,pot}} = \frac{\rho}{\rho_{\mathrm{pot}}} U_{\mathrm{tot}} \tag{30c}$$

using the gas constant of air $R_{\mathrm{d}}$ and the specific heat capacity at constant pressure $c_{\mathrm{p}}$. These potential variables are free of those pressure fluctuations near the engine exit (Lewellen, 2020). The potential temperature is the temperature of an air parcel that results when it is adiabatically compressed/expanded to ambient pressure (in atmospheric physics, typically the surface pressure is used as reference, here $p_{\mathrm{a}}$ is the reference pressure). The definition of potential density follows from the ideal gas law. By using potential velocity values, the mass flows are maintained. These potential variables reduce to the actual temperature $T$, density $\rho$, and velocity $U_{\mathrm{tot}}$ when pressure equals the ambient pressure $p_{\mathrm{a}}$.

With these potential variables, we evaluate the integrated total enthalpy flow

$$\dot{E}_{\mathrm{tot}}(x) = \dot{E}_{\mathrm{static}}(x) + \dot{E}_{\mathrm{kin}}(x) \\ = \sum_{i} \left[ c_{\mathrm{p}}(T_{\mathrm{pot,i}} - T_{\mathrm{a}}) + \frac{1}{2}(U_{\mathrm{tot,pot,i}} - U_{\infty})^2 \right] \rho_{\mathrm{pot,i}} A_i U_{\mathrm{tot,pot,i}} \ , \tag{31}$$

where $i$ goes over all grid boxes of interest with cross-sectional area $A_i$ at a certain downstream distance $x$. Evaluation of Eq. (31) over the whole exhaust, i.e., grid boxes where fractions of bypass and/or core air are contained, confirms that kinetic energy is dissipated into heat, yet the total enthalpy flow is a conserved quantity (Fig. 2b). The core plume enthalpy flow (grid boxes with core air fractions), however, increases with downstream distance until the bypass energy is fully entrained into the plume at $x \approx 60\,\mathrm{m}$. Moreover, the kinetic energy flow within the core plume increases within the first $x \approx 40\,\mathrm{m}$ meaning that the entrainment of kinetic energy from the bypass flow into the plume exceeds the viscous dissipation within the plume. From the total enthalpy flows $\dot{E}_{\mathrm{tot}}$ and $\dot{E}_{\mathrm{tot,plume}}$ near the engine exit, we estimate $\beta \approx 0.62$. Furthermore, we estimate the core exit temperature $T_{\mathrm{pot,E}} \approx 564\,\mathrm{K}$, the total core jet velocity $U_{\mathrm{tot,pot,E}} \approx 428\,\mathrm{m\,s^{-1}}$ and core radius $r_{\mathrm{E}} \approx 0.33\,\mathrm{m}$. Moreover, we





estimate the bypass ratio $b$ from the ratio of the fractions $\mathcal{B}$ and $\mathcal{D}$ far downstream of the engine when core and bypass air are

well-mixed. We use the estimated value as a normalization constant in the definition of $\gamma$ (Eq. (24)).

We calculate the average of a mass-specific quantity $\xi$ (tracer mass fraction, momentum, static enthalpy, kinetic energy, ...) as mass flow weighted average over the core plume at a certain downstream distance $x$, i.e.,

$$\overline{\xi} = \frac{\sum_i \rho_{\text{pot},i} A_i U_{\text{tot,pot},i} \xi_i}{\sum_i \rho_{\text{pot},i} A_i U_{\text{tot,pot},i}} \ . \tag{32}$$

This type of averaging ensures that the (tracer) mass flows and energy flows calculated with average values correspond to the

total flows in the 3D field. Generally, $i$ goes over all grid boxes involving core air, i.e., $\mathcal{D}_i > 0$. However, the physical diffusion processes are partially obscured by numerical diffusion. Therefore, we define at each downstream location $x$ a relative threshold $\delta$ for the plume boundary and consider the set of grid boxes that fulfill

$$\{i \mid \mathcal{D}_i(x) > \delta \cdot \max(\mathcal{D}(x))\} \ . \tag{33}$$

Per default, we set $\delta = 0.01$, but we also tested the impact of other choices (Sec. 4.5).

It is not necessarily true, that $\overline{E}_{\text{kin}} = \frac{1}{2}(\overline{U}_{\text{tot,pot}} - U_\infty)^2$. Here, $\overline{U}_{\text{tot,pot}}$ is obtained by substituting $\xi_i = U_{\text{tot,pot},i}$ into Eq. (32). Instead, the correct calculation of the average kinetic energy $\overline{E}_{\text{kin}}$ is done by setting $\xi_i = \frac{1}{2}(U_{\text{tot,pot},i} - U_\infty)^2$ in Eq. (32). These correctly calculated $\overline{E}_{\text{kin}}$ values have to be inserted into Eq. (28) when an average plume trajectory is considered. For such an average plume trajectory, we convert a downstream distance $x$ to an average plume age

$$t(x) = \int_0^x \mathrm{d}\tilde{x}/\overline{U}_{\text{tot,pot}}(\tilde{x}) \ . \tag{34}$$

Core and bypass air mix near the engine exit (Fig. 2c). In the analysed CFD data, the core air fraction $\overline{\mathcal{D}}$ is initially already less than one, and the bypass air fraction $\overline{\mathcal{B}}$ larger than zero, as the first data slice is available at a downstream position slightly behind the core engine exit. In the first meters behind the engine exit, core air dilution is primarily driven by the shear between the core and bypass flow. However, before the bypass fraction $\overline{\mathcal{B}}$ reaches its maximum at $\approx 30\,\mathrm{m}$, ambient air is already contained in the plume and a noticeable increase in the dilution speed of the core air $\overline{\mathcal{D}}$ is visible. This change in dilution speed

indicates that the shear resulting from the velocity difference between the core plume and the ambient air is becoming more relevant. As this change in dilution speed occurs before the onset of microphysics, the initially reduced shear between the core and bypass flows may not have a significant influence on the contrail formation process - at least in our bulk approach with an average trajectory. However, as the bypass air is not fully contained in the core plume at the start of contrail formation ($\overline{\gamma}$ is less than one in Fig. 2d), the thermodynamics (Eq. (28)) may have an impact.

The average temperature evolutions agree when calculated in two ways (Fig. 2e): directly from the 3D field by setting $\xi_i = T_{\text{pot},i}$ in Eq. (32) and by evaluating Eq. (27) inserting average quantities $\overline{E}_{\text{kin}}$, $\overline{\mathcal{D}}$ and $\overline{\gamma}$. This justifies the approach and its assumptions outlined in Sec. 2.4.1, in particular the assumptions of a constant $c_{\text{p}}$ and the same dilution factors $\mathcal{D}$/entrainment rates $\omega$ for enthalpy and mass. The initially higher mean plume temperature for a stagnant plume with the emitted combustion heat assumed to be contained in the core flow at the engine exit (Fig. 2e), leads to lower mean plume relative humidity values



compared to the case with complete treatment of kinetic and bypass energy (Fig. 2f). The temperature and relative humidity evolutions converge when the bypass and core air are mixed, and kinetic energy has dissipated into heat.

## 3  Box model setup

We use the Lagrangian Cloud Module (LCM) in the box model version described in detail in Bier et al. (2024) and extended by the various aspects outlined in Sec. 2. In the particle-based LCM box model, aerosol particles and hydrometeors (droplets

and ice crystals) are represented by simulation particles (SIPs). The time-resolved microphysical processes on these SIPs are simulated offline without feedback on the dynamics. Instead, dilution is prescribed analytically or using data from a previously performed CFD simulation. We incorporated dilution data from multiple sources into the box model, each briefly summarized in this section. The ambient and engine exit conditions under which the different CFD datasets were produced are provided in Table 2.

**FLUDILES:** The dilution is derived from a LES conducted for a single uninstalled CFM56 engine representative of an A340-300 aircraft. From this LES, Vancassel et al. (2014) generated an ensemble of 25000 trajectories, each representing a fraction of the plume's mass/volume. Bier et al. (2024) then used a suitably merged subset of 1000 trajectories for their box model simulations. In our simulations, we utilize both this 1000-trajectory ensemble and a single average trajectory, which represents the mass-weighted average of the ensemble (see Supplement in Bier et al., 2022). The dilution factor is determined from the

data by treating the temperature as a passive tracer. In the trajectory ensemble approach, box model simulations are performed for each member and results are presented as sum over all trajectories for extensive variables (e.g., ice crystal number) and as mass-weighted average for intensive variables (e.g., temperature). For the average trajectory approach, a single box model simulation is performed.

**Lewellen:** We use the average dilution factor obtained from a temporal LES simulation for an isolated engine representative of

a B737 aircraft (Lewellen, 2020). The dilution factor is determined by an exhaust-weighted average (see Sec. 2d in Lewellen (2020) for details). These dilution data have previously been used alternatively to the FLUDILES data in Bier et al. (2022).

**Airbus Turbofan:** We use the average dilution factor obtained from the RANS data we obtained from Airbus. In addition to the average dilution factor $\overline{\mathcal{D}}$ we also incorporated the extracted $\beta$, $\overline{\gamma}$ and $\overline{E}_{\mathrm{kin}}$. Details have been described in Sec. 2.4.2. The engine is designed for a Short-Medium Range (SMR) scenario (a A320 like aircraft).

**Analytical Kärcher formula:** We use the analytical formula presented in Kärcher et al. (2015), which reads

$$\mathcal{D}(t > \tau_{\mathrm{m}}) = \left( \frac{\tau_{\mathrm{m}}}{t} \right)^{B} \tag{35}$$

and $\mathcal{D} = 1$ for $t \leq \tau_{\mathrm{m}}$ with $\tau_{\mathrm{m}} \approx 0.01\,\mathrm{s}$. The exponent $B$ determines the dilution speed with a default value set to $B = 0.9$ after Kärcher (1999). Setting $B = 1.15$ leads to a dilution evolution more comparable to the other data sources (Fig. 3a).

The time series of $\mathcal{D}$ differ substantially between the various CFD data sources (Fig. 3a). Consistent with our theoretical

understanding, the dilution data show a much smaller spread after the dilution time scaling based on the engine size and jet speed (Eq. (19)) is applied (Fig. 3b).





**Table 2.** Ambient temperature $T_a$, ambient pressure $p_a$, flight velocity $U_\infty$, core engine exit excess jet velocity $U_{\text{jet,E}}$, core engine exit temperature $T_E$ and core exit radius $r_E$ for which the various CFD data were produced.

| CFD data | $T_a$ / K | $p_a$ / hPa | $U_\infty$ / m s$^{-1}$ | $U_{\text{jet,E}}$ / m s$^{-1}$ | $T_E$ / K | $r_E$ / m |
|---|---|---|---|---|---|---|
| **FLUDILES** | 220 | 240 | 250 (Mach 0.84) | 230 | 580 | 0.50 |
| **Lewellen** | 218.8 | 238.4 | 237 (Mach 0.80) | $\approx 245$ | $\approx 580$ | $\approx 0.30$ |
| **Airbus Turbofan** | 218.8 | 238.4 | 231 (Mach 0.78) | $\approx 197$ | $\approx 564$ | $\approx 0.33$ |

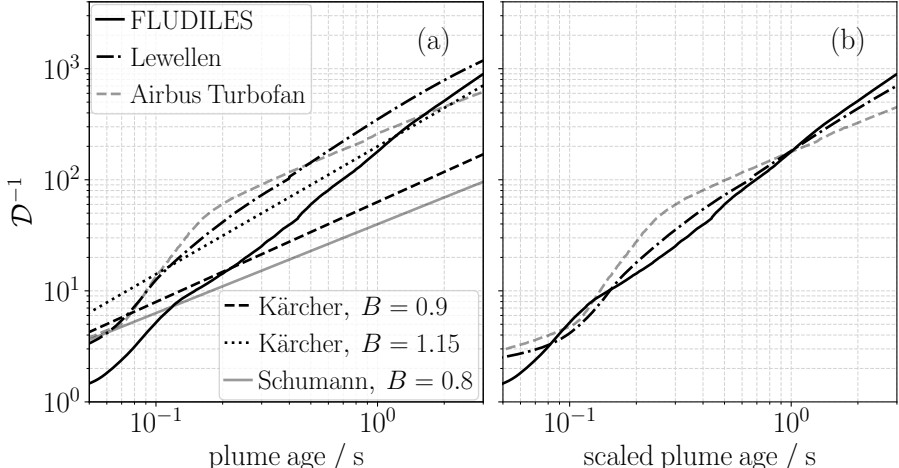

**Figure 3.** (a) Inverse dilution factor for average trajectories derived from different data sources. For the Airbus Turbofan, the dashed line represents the case where $\delta = 0.01$ is set in Eq. (33). The boundaries of the shaded region correspond to $\delta = 0.05$ and $\delta = 0.002$, respectively. The solid gray line is calculated with the Kärcher formula (Eq. (35)) with an exponent $B = 0.8$ as proposed by Schumann et al. (1998). (b) Inverse dilution factors after scaling the core exit radius and exit velocity to the values of the FLUDILES engine (Table 2) with associated dilution time scaling (Eq. (19)).

In the result section (Sec. 4) the box model initializations differ depending on the purpose of each section. Therefore, we briefly summarize the settings at the beginning of each subsection. When not stated differently, we use the FLUDILES trajectory ensemble as dilution data and the baseline values listed in Table 3 as model setup.

## 4 Results

In this section, we analyze how the different aspects described in Sec. 2 affect the contrail formation on ambient aerosols for hydrogen combustion. Specifically, we look at their impact on the final number of ice crystals formed $N_{\text{ice,f}}$, which is the ice crystal number when contrail formation is finished. Furthermore, we develop appropriate scaling relations of $N_{\text{ice,f}}$. These scaling relations allow to describe the sensitivity to selected parameters by simple analytical expressions, which are easily incorporated in the $N_{\text{ice,f}}$ parametrization.





**Table 3.** Baseline values of ambient pressure $p_a^*$, ambient relative humidity $RH_{ice,a}^*$, aircraft speed $U_\infty^*$, water vapor emission index $EI_v^*$, specific combustion heat $Q^*$, overall propulsion efficiency $\eta^*$, exit temperature $T_E^*$, exit area $A_E^*$, exit excess jet velocity $U_{jet,E}^*$, geometric mean radius of aerosol particles $\overline{r}_{d,aer}^*$, geometric width $\sigma_{aer}^*$, hygroscopicity $\kappa_{aer}^*$ and number concentration $n_{aer}^*$.

| | |
|---|---|
| background conditions | $p_a^* = 260\,\text{hPa}, RH_{ice,a}^* = 115\,\%, U_\infty^* = 250\,\text{m s}^{-1}$ |
| fuel/engine properties | $EI_v^* = 8.94\,\text{kg kg}^{-1}, Q^* = 120\,\text{MJ kg}^{-1}, \eta^* = 0.4$ |
| engine exit conditions | $T_E^* = 580\,\text{K}, A_E^* = 0.25\pi\,\text{m}^2, U_{jet,E}^* = 230\,\text{m s}^{-1}$ |
| aerosol properties | $\overline{r}_{d,aer}^* = 20\,\text{nm}, \sigma_{aer}^* = 1.6, \kappa_{aer}^* = 0.5, n_{aer}^* = 1000\,\text{cm}^{-3}$ |

## 4.1 Impact of engine exit conditions for changing ambient temperature on $N_{ice,f}$

In Sec. 2.3.1, we presented scaling relations describing how engine exit conditions and dilution speed vary with ambient temperature and pressure changes. The linear scaling of density and fuel flow with ambient pressure (Table 1 and Eq. (10)) has already been consistently applied in previous box model simulations by Bier et al. (2022, 2024). In these studies, however, they
held the engine exit temperature $T_E$ constant and did not adjust the dilution speed when they changed the ambient temperature. Here, we investigate whether these neglects significantly affect $N_{ice,f}$.

The FLUDILES data were produced with an ambient temperature $T_a^* = 220\,\text{K}$ and an exit temperature $T_E^* = 580\,\text{K}$ (Table 2). If the ambient temperature $T_a = 210\,\text{K}$ is specified, the scaling suggested in Sec. 2.3.1 results in $T_E \approx 554\,\text{K}$ and $s_{dil} \approx 1.02$. For $T_a = 232\,\text{K}$ these two values are $T_E \approx 612\,\text{K}$ and $s_{dil} \approx 0.97$. However, the difference in the number of ice crystals formed
is negligible between simulation runs with the scalings applied and simulation runs with fixed $T_E = T_E^* = 580\,\text{K}$ and without dilution time scaling (Fig. 4). Therefore, we can safely neglect these scaling relations in subsequent analyses and instead prescribe the same exit temperature and the same unscaled dilution regardless of the actual ambient temperature.

## 4.2 Scaling relation for $N_{ice,f}$ for variation of overall propulsion efficiency

This subsection analyses how $N_{ice,f}$ changes with variations of the overall propulsion efficiency $\eta$ and how $N_{ice,f}(\eta)$ can be
expressed in terms of $N_{ice,f}(\eta^*)$ for a fixed reference value $\eta^*$. In a first step, we perform sensitivity studies with the FLUDILES trajectory ensemble, where a variation of $\eta$ is reflected by corresponding adaptations of the engine exit conditions (as outlined in Sec. 2.2). From theoretical considerations, a higher $\eta$ value implies a steeper mixing line and higher and longer-lasting supersaturation. Consistent with this, Fig. 5 shows increasing ice crystal numbers $N_{ice,f}$ with increasing $\eta$. Clearly, $N_{ice,f}$ itself and the change of it with $\eta$ depend on the meteorological background conditions. In situations with high plume supersaturations
and large $N_{ice,f}$ values (i.e., lower ambient temperature and higher ambient pressure), a change in $\eta$ has less pronounced effect on $N_{ice,f}$. For an higher ambient temperature ($T_a = 232\,\text{K}$), however, an increase in $\eta$ leads to a noticeable increase in $N_{ice,f}$.



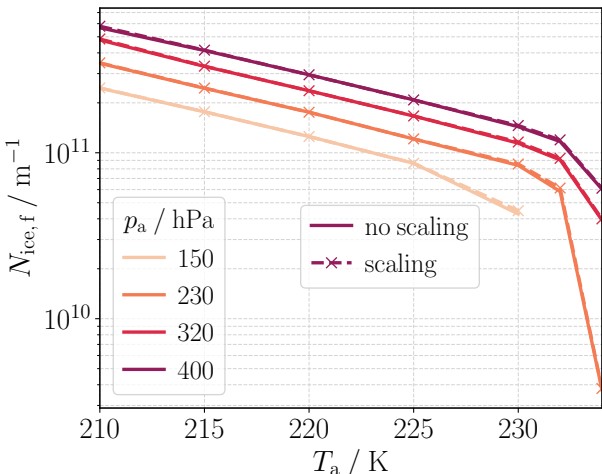

**Figure 4.** Impact of engine exit conditions for varying ambient temperature $T_a$ on the number of ice crystals formed $N_{ice,f}$. Simulation results for scaled engine exit temperature (Table 1) and scaled dilution time (Eq. (21)) with $T_a^* = 220\,$K are compared to simulations with fixed exit temperature $T_E = T_E^* = 580\,$K and without dilution time scaling applied. Results are shown for $N_{ice,f} > 0$.

The slope $G$ of the mixing line depends on both, the overall propulsion efficiency and the ambient pressure (Eq. (7)). The value pair $\eta$ and $p_a$ corresponds to the the same slope as the value pair $\eta^*$ and $\tilde{p}_a$ with the definition

$$\tilde{p}_a = p_a \frac{1 - \eta^*}{1 - \eta} \quad . \tag{36}$$

Therefore, we try to mimic the effect of a variation of the overall propulsion efficiency by fixing its value to $\eta^* = 0.4$ but adjusting the ambient pressure after Eq. (36). In the box model setup of this simulation series with $\eta^* = 0.4$, we fix the exit temperature to $T_E^* = 580\,$K, neglecting the impact of the ambient temperature on the exit condition (Sec. 4.1) while maintaining consistency with the scaling relations presented in Sec. 2.2. Consequently, the exit conditions for the tuple $(\eta, p_a)$ are different from the tuple $(\eta^*, \tilde{p}_a)$, meaning the two approaches start from different points on the mixing line. For an ambient temperature

$T_a = 225\,$K, the engine exit temperature is $T_E = 665\,$K for $\eta = 0.35$, and $T_E = 514\,$K for $\eta = 0.45$ using Eq. (15).

For the aerosol number concentration $n_{aer} = 100\,$cm$^{-3}$, the approximation with $\eta^*$ and $\tilde{p}_a$ works almost perfectly (Fig. 5a). For $n_{aer} = 1000\,$cm$^{-3}$, the approximation slightly overestimates the ice crystal number for $\eta$ values above and slightly underestimates it for values below $\eta^* = 0.4$ (Fig. 5b). The deviations come from the fact that the ambient pressure goes into other calculations within the box model (e.g., mean free path of air molecules, diffusion coefficient of water vapor) and that the

starting points on the mixing line are different. However, as the deviations are only marginal, this suggests a low sensitivity of $N_{ice,f}$ on the exact engine exit conditions.

In summary, changes in overall propulsion efficiency can be well approximated by adjusting the ambient pressure. Therefore, when developing a parameterization for the number of ice crystals formed, it will be sufficient to represent the pressure dependence accurately.





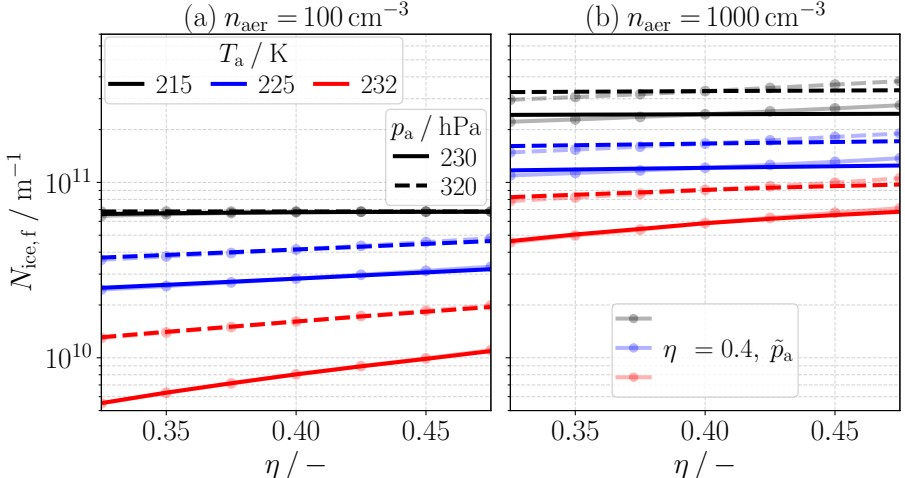

**Figure 5.** Influence of a variation of overall propulsion efficiency $\eta$ on the number of ice crystals formed $N_{\mathrm{ice,f}}$ for three ambient temperatures $T_{\mathrm{a}}$ (different color), two ambient pressures $p_{\mathrm{a}}$ (different line styles) and two ambient aerosol number concentrations $n_{\mathrm{aer}}$ (different panel). Transparent lines show simulation results for fixed overall propulsion efficiency $\eta^* = 0.4$ but adjusted ambient pressure $\tilde{p}_{\mathrm{a}}$ after Eq. (36).

### 4.3 Scaling relations for $N_{\mathrm{ice,f}}$ for different engine sizes

In the following, we investigate the effect of the engine size $r_{\mathrm{E}}$ on contrail formation and propose scaling relations between engine size and the number of ice crystals formed $N_{\mathrm{ice,f}}$. To do so, we perform box model simulations based on the FLUDILES trajectory ensemble. While keeping all other parameters fixed, the core exit radius $r_{\mathrm{E}}$ is scaled by the factor $s_{\mathrm{E}}$ (see definition in Eq. (20b)). According to Sec. 2.3, this involves also a change of the dilution time scale by a factor $s_{\mathrm{dil}}$ (Eq. (19)). While the scaling factors $s_{\mathrm{E}}$ and $s_{\mathrm{dil}}$ refer to different aspects of the problem, in our particular case, where only $r_{\mathrm{E}}$ is varied and $s_{\mathrm{shear}} = 1$, $s_{\mathrm{dil}}$ takes the value of $s_{\mathrm{E}}$.

The (effective) plume area at engine exit scales as $\hat{A}_{\mathrm{E}} \propto s_{\mathrm{E}}^2$. If the engine size variation had only this trivial geometric effect and the plume dilution time scale had no effect on contrail formation, then a first-order estimate would give $N_{\mathrm{ice,f}} \propto \hat{A}_{\mathrm{E}} \propto s_{\mathrm{E}}^2$. This first-order estimate holds, if the normalized ice crystal numbers $N_{\mathrm{ice,f}}/s_{\mathrm{E}}^2$ are independent of $s_{\mathrm{E}}$ and lie on a universal curve for different $s_{\mathrm{E}}$ values. Fig. 6a reveals that this is approximately the case for low aerosol number concentrations $n_{\mathrm{aer}}$ at ambient temperatures $T_{\mathrm{a}} \lesssim 230\,\mathrm{K}$. However, the curves corresponding to the different $s_{\mathrm{E}}$ values start to deviate at higher aerosol number concentrations. A slower dilution leads to lower (scaled) $N_{\mathrm{ice,f}}$ values because aerosol particles that are entrained and activated into ice crystals at an early stage have more time to consume the water vapor. While these early-activated ice crystals grow, they reduce the supersaturation, preventing later-entrained aerosols from being activated into ice crystals.

Lewellen (2020) derived a scaling relation by considering the time interval during which a plume parcel can potentially activate aerosols and the time interval required for ice crystals to grow large enough to deplete the available water vapor



significantly. In our application, this scaling is expressed as (see Appendix B)

$$\frac{N_{\text{ice,f}}}{s_{\text{E}}^2 \cdot s_{\text{dil}}^a} \sim f(s_{\text{dil}}^{3/2} \cdot n_{\text{aer}}) \tag{37}$$

with $a = -3/2$ and $f$ being a function that depends on ambient conditions and aerosol properties but not explicitly on the engine size/dilution speed. This means that the curves for different $s_{\text{E}}$ values should collapse when the scaling is applied. Indeed, this scaling approach works reasonably well for temperatures $T_{\text{a}} \lesssim 230\,\text{K}$ (Fig. 6b). However, for $T_{\text{a}} = 233\,\text{K}$, the curves for the different $s_{\text{E}}$ values do not collapse, hinting that another timescale might become important at high ambient temperatures (which is not relevant in kerosene combustion scenarios, for which this scaling was originally developed).

We will extend the existing scaling approach by considering the limiting factor of droplet freezing. Aerosols that are activated into water droplets have to grow large enough before the freezing process is triggered (Bier et al., 2024). The duration between droplet activation and freezing into ice crystals depends on the time point at which an aerosol particle is entrained into the plume, specifically on the prevailing plume temperature, plume supersaturation, and their subsequent evolution. As a first-order estimate, we use the time difference between the first droplet activations and the first ice crystal formations in our simulations as a proxy for the freezing timescale $\delta t_{\text{frz}}$ (Tab. 4). We compare this timescale $\delta t_{\text{frz}}$ to the maximum possible duration of water supersaturation $\delta t_{\text{supersat}}$. In the absence of microphysical processes, the latter depends on dilution speed and scales linearly with $s_{\text{dil}}$. In Table 4, we present the average of $\delta t_{\text{supersat}}$ over all trajectories for a prescribed value of $s_{\text{dil}} = 1$.

At low ambient temperatures, supersaturations are high and long-lasting. With increasing $T_{\text{a}}$, however, $\delta t_{\text{supersat}}$ decreases. In contrast, $\delta t_{\text{frz}}$ is short at low ambient temperatures and increases with increasing ambient temperature. Hence, the relative importance of $\delta t_{\text{frz}}$ grows with increasing temperature. This means that, at high ambient temperatures, a substantial portion of aerosols that are entrained and activated into droplets at a later stage are unable to freeze before the relative humidity over water falls below $100\,\%$. These droplets evaporate subsequently. Clearly, this is more pronounced for faster dilution, i.e., shorter $\delta t_{\text{supersat}}$, which explains the relative order of the three curves for $T_{\text{a}} = 233\,\text{K}$ in Fig. 6b.

However, these three curves are shifted by an approximately constant factor. This allows the freezing time scale to be accounted for by adapting the exponent $a$ in Eq. (37). We propose maintaining $a = -3/2$ up to $T_{\text{a}} = 230\,\text{K}$, followed by a linear increase with ambient temperature, specifically,

$$a = \begin{cases} -\frac{3}{2} & \text{for } T_{\text{a}} \leq 230\,\text{K} \\ \frac{1}{6}\left(T_{\text{a}}/\text{K} - 239\right) & \text{for } 230\,\text{K} < T_{\text{a}} \leq 235\,\text{K} . \end{cases} \tag{38}$$

This adapted scaling approach performs reasonably for temperatures $T_{\text{a}} \lesssim 233\,\text{K}$ (Fig. 6c). For $T_{\text{a}} = 234\,\text{K}$, however, the symmetry breaks between the decelerated dilution ($s_{\text{dil}} = s_{\text{E}} = 2$) and the accelerated dilution ($s_{\text{dil}} = s_{\text{E}} = 0.5$). Whereas for $s_{\text{dil}} = 2$ the scaling still works reasonably well, the (scaled) $N_{\text{ice,f}}$ values are lower for $s_{\text{dil}} = 0.5$. This means that the function $f$ introduced in Eq. (37) becomes dependent on the dilution speed, thus limiting the applicability of the scaling approach above $T_{\text{a}} \sim 233\,\text{K}$.



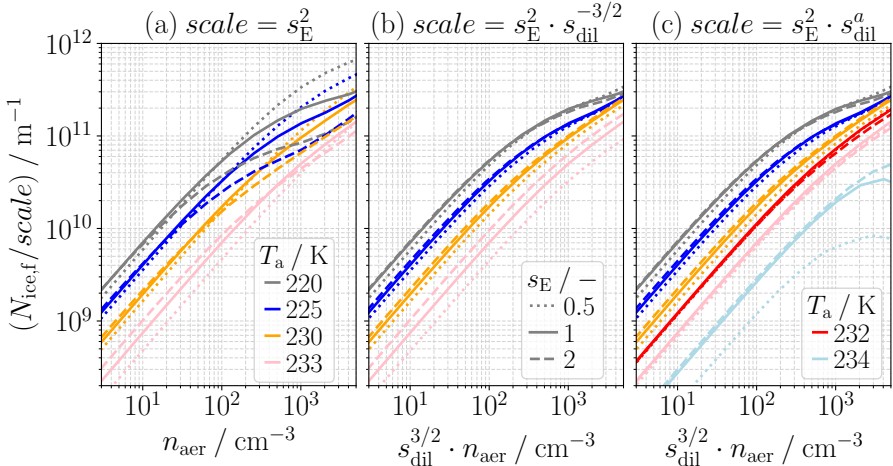

**Figure 6.** Scaling relations for the number of ice crystals formed $N_{ice,f}$ for different engine sizes $s_E$ (different line styles). Shown are scaled $N_{ice,f}$ values as a function of (scaled) aerosol number concentration $n_{aer}$ for different ambient temperatures $T_a$ (different colors). (a) Normalization by scale of engine exit area. (b) Scaling after Lewellen (2020) with $s_{dil} = s_E$. (c) Adapted scaling with variable exponent $a$ after Eq. (38).

**Table 4.** Average duration of water supersaturation $\delta t_{supersat}$ without microphysics and proxy for freezing timescale $\delta t_{frz}$ for different ambient temperatures $T_a$ for the FLUDILES trajectory data set with $s_{dil} = 1$.

| $T_a / K$ | 220 | 225 | 230 | 231 | 232 | 233 |
|---|---|---|---|---|---|---|
| $\delta t_{supersat} / s$ | 6.5 | 2.3 | 1.4 | 1.2 | 1.1 | 1.0 |
| $\delta t_{frz} / s$ | < 0.1 | < 0.1 | 0.1 − 0.2 | 0.1 − 0.2 | 0.3 − 0.4 | 0.4 − 0.5 |

## 4.4 Impact of different dilution data on $N_{ice,f}$

In Sec. 3, we described several data sets to prescribe the dilution in the box model. In the following, we investigate the influence
of the dilution data on our model results.

In a first step, we compare results for the FLUDILES trajectory ensemble and the average trajectory (as already done by Bier et al. (2022) for contrail formation on soot particles for kerosene combustion). Both approaches yield a comparable number of ice crystals $N_{ice,f}$ when the ambient temperature $T_a$ is below $\sim 230\,K$ (Fig. 7). At higher temperatures, however, $N_{ice,f}$ values for the ensemble approach tend to be higher than for the average trajectory approach. Furthermore, scenarios exist where no ice
crystals form in the average trajectory, whereas $N_{ice,f}$ is larger than zero in the ensemble approach. How large the discrepancies between the two approaches are depends on the peak plume supersaturation (e.g., affected by ambient pressure, Fig. 7a-b) and on the aerosol properties (Fig. 7a,c,d). The key factor in understanding these differences is the interplay between the timescale of supersaturation $\delta t_{supersat}$ and freezing $\delta t_{frz}$ (Sec. 4.3). In the ensemble approach, the 1000 members dilute at different




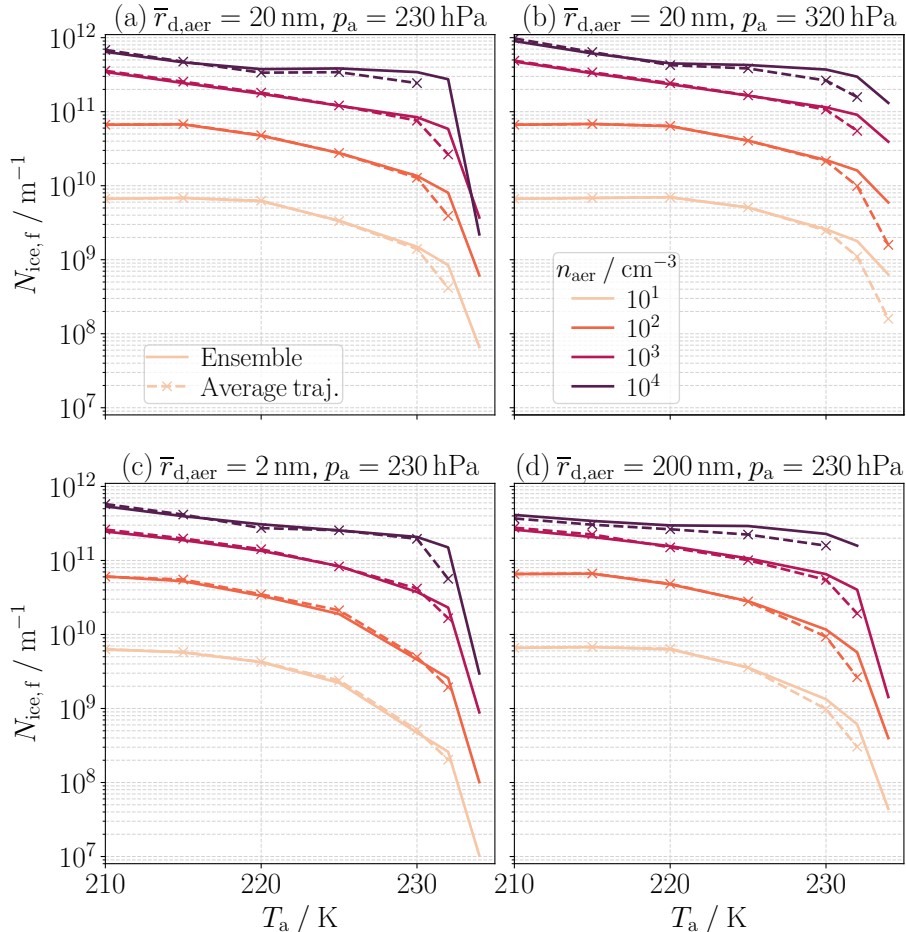

**Figure 7.** Comparison of FLUDILES ensemble (solid) and average trajectory (dashed) approaches. Shown are the number of ice crystals formed $N_{\mathrm{ice,f}}$ as a function of ambient temperature $T_{\mathrm{a}}$ for different aerosol number concentrations $n_{\mathrm{aer}}$ (different colors). Each panel depicts results for a different combination of aerosol geometric mean radius $\overline{r}_{\mathrm{d,aer}}$ and ambient pressure $p_{\mathrm{a}}$. Results are shown for $N_{\mathrm{ice,f}} > 0$.

speeds. Some trajectories dilute more slowly, allowing more droplets to grow sufficiently to freeze into ice crystals. In contrast,
the freezing is more of a binary process in the single average trajectory approach, resulting in lower or zero ice crystal number at the same temperature.

The different dilution data were produced for different engine characteristics and background conditions (Table 2). However, we only extract the information on dilution (in terms of $\mathcal{D}$ and $\omega$) and the information on engine size and jet speed in the subsequent analysis. Particularly, we still rely on classical contrail formation theory and do not consider the thermodynamic
influence of the bypass flow and kinetic energy dissipation. This will be investigated in Sec. 4.5. In each simulation setup we use the same exit temperature $T_{\mathrm{E}} = 580\,\mathrm{K}$ and overall propulsion efficiency $\eta = 0.4$, obviously neglecting differences among



the engines. However, it disentangles the impact of those values from the influence of the dilution on $N_{\mathrm{ice,f}}$. Moreover, we intend to use the theoretically derived dilution time scaling (Sec. 2.3) with associated scaling of $N_{\mathrm{ice,f}}$ (Eq. (37)) neglecting any further details in the configuration of the engines.

Since the number of entrained particles (Eq. (12)) and thus the number of ice crystals formed is proportional to the effective core exit area $\hat{A}_{\mathrm{E}}$ (Eq. (11)), we use the geometric scale $\hat{s}_{\mathrm{E}}^2 = \hat{A}_{\mathrm{E}}/\hat{A}_{\mathrm{E}}^*$ to make the results for the different-sized engines comparable. We use the FLUDILES engine as reference, thus $\hat{A}_{\mathrm{E}}^*$ refers to its effective core area. Relating the effective core areas instead of the physical core areas (as done with the scale $s_{\mathrm{E}}^2$ in Sec. 4.3) additionally accounts for the different ratios of total jet and aircraft velocities in the CFD data. The dilution time scale $s_{\mathrm{dil}}$, however, is still determined with Eq. (19) using $s_{\mathrm{E}}$

and $s_{\mathrm{shear}}$. As the FLUDILES engine is treated as reference, the associated values are $\hat{s}_{\mathrm{E}} = s_{\mathrm{dil}} = 1$ for this engine. Moreover, we set $\hat{s}_{\mathrm{E}} = s_{\mathrm{dil}} = 1$ for the Kärcher dilution as the analytical formula does not contain an explicit dependency on engine size or jet speed. Instead, the dilution speed is governed by the exponent $B$ in Eq. (35).

Simulations using the default Kärcher dilution ($B = 0.9$ in Eq. (35)), yield significantly different ice crystals numbers compared to those using the FLUDILES dilution (Fig. 8a). This difference arises from the markedly slower dilution speed (Fig. 3).

However, when a faster dilution with $B = 1.15$ is used (Fig. 8b), the resulting ice crystal numbers are similar to those from the FLUDILES simulations. This suggests that the exact exact shape of the dilution curve over time (Fig. 3) is of lesser importance for the final number of ice crystals formed.

The faster Lewellen dilution (Fig. 3) caused by the smaller core radius (Table 2), leads to higher (scaled) $N_{\mathrm{ice,f}}$ values at high aerosol number concentrations (Fig. 8c). When the scaling after Eq. (37) is applied, the resulting curves for a given ambient

temperature align reasonably well (Fig. 8d), demonstrating the consistency of the scaling approach across different dilution scenarios.

The scaling performs somewhat less ideally for the Turbofan dilution data (Fig. 8f). Especially for $T_{\mathrm{a}} = 215\,\mathrm{K}$, the scaled curves have slightly different shapes. This may be partially attributed to the dynamic influence of the bypass flow on the dilution speed. Additionally, droplets do not freeze in the Airbus Turbofan case for $n_{\mathrm{aer}} \gtrsim 1000\,\mathrm{cm}^{-3}$ at $T_{\mathrm{a}} = 232\,\mathrm{K}$ in contrast to the

FLUDILES simulations. Nevertheless, the scaling after Eq. (37) based on core exit conditions still accounts for the leading impact of the dilution speed on $N_{\mathrm{ice,f}}$. Moreover, $N_{\mathrm{ice,f}}$ is only weakly sensitive on the exact choice of the plume boundary in Eq. (33) (Fig. 8e-f).

## 4.5 Impact of bypass flow and kinetic energy dissipation on $N_{\mathrm{ice,f}}$

Using the Airbus Turbofan dilution data, we investigate the impact on the number of ice crystals formed for the two approaches

outlined in Sec. 2.4.1. The first approach explicitly considers the thermodynamics of the bypass flow, including kinetic energy effects. The second, simplified approach assumes that the whole emitted combustion heat is contained as static enthalpy in the core flow at the engine exit. In the former approach we prescribe the estimated exit conditions for the Airbus Turbofan data (Table 2 and $\beta = 0.62$) and use the extracted time evolutions of $\overline{\mathcal{D}}$, $\overline{E}_{\mathrm{kin}}$ and $\overline{\gamma}$ (Sec. 2.4.2). In the latter approach we make use of only $\overline{\mathcal{D}}$ and prescribe the adjusted exit temperature $\tilde{T}_{\mathrm{E}}$ after Eq. (29). Using the same $T_{\mathrm{E}} = 564\,\mathrm{K}$ for all ambient

temperatures leads to $\tilde{T}_{\mathrm{E}} = 809\,\mathrm{K}$ for $T_{\mathrm{a}} = 215\,\mathrm{K}$ and $\tilde{T}_{\mathrm{E}} = 798\,\mathrm{K}$ for $T_{\mathrm{a}} = 232\,\mathrm{K}$.





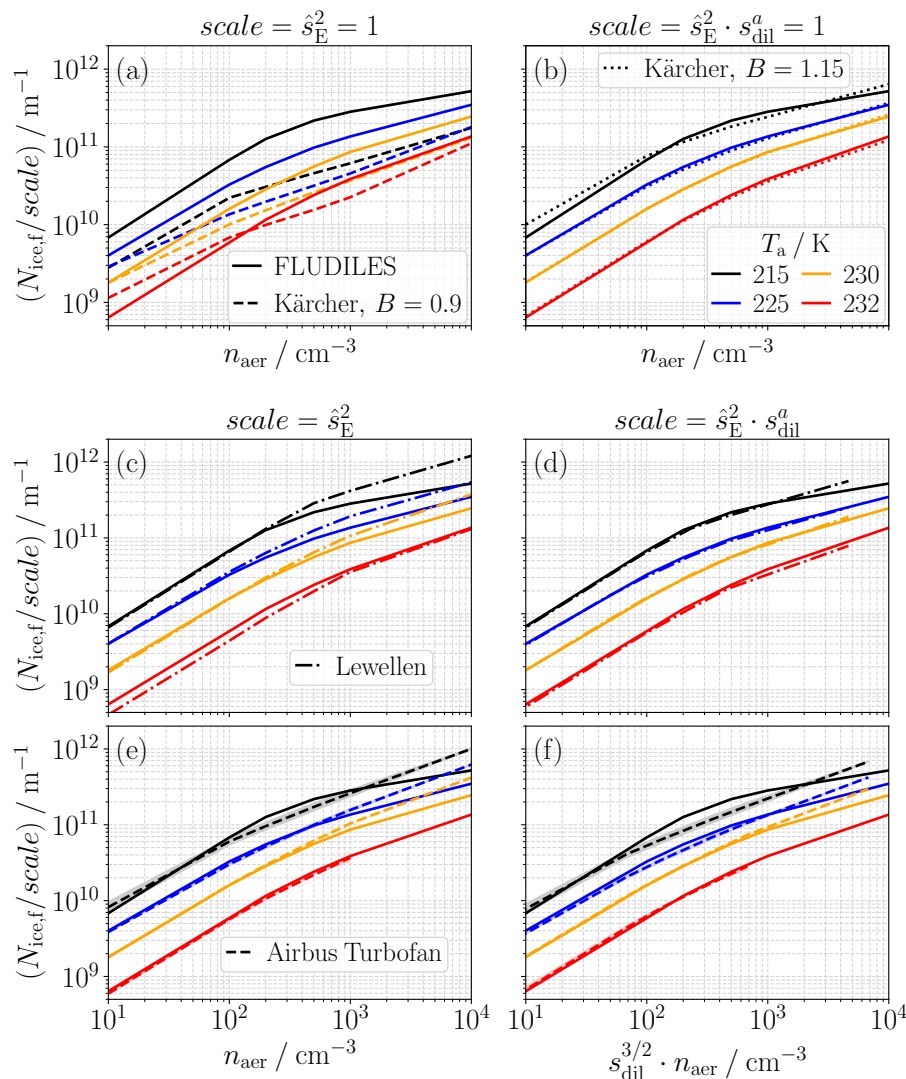

**Figure 8.** Impact of different dilution data on the number of ice crystals formed $N_{ice,f}$. In the left column, $N_{ice,f}$ is normalized by the scale of the effective engine exit area $\hat{s}_E^2$. In the right column, the scaling after Eqs. (37) and (38) is applied. For the FLUDILES and the Kärcher dilution, it holds $\hat{s}_E = s_{dil} = 1$. In each panel, $N_{ice,f}$ is plotted for the FLUDILES average trajectory (solid lines) as a function of (scaled) aerosol number concentration $n_{aer}$ for different ambient temperatures $T_a$ (different colors). In each row, results for a second dilution type (indicated by the line style) are compared to the FLUDILES results. For the Airbus Turbofan (panel (e)-(f)), the dashed line shows the simulation results where $\delta = 0.01$ is set in Eq. (33). The boundaries of the shaded regions correspond to $\delta = 0.05$ and $\delta = 0.002$, respectively. Results are shown for $N_{ice,f} > 0$.

The former approach leads to initially higher plume supersaturation (Fig. 9a) and thus to a slightly earlier onset of droplet activation (Fig. 9b). However, this has little influence on the final number of ice crystals formed (Fig. 9b–d). This insignificant





impact is due to the continuous entrainment of ambient aerosols into the core plume over time. Since we assume that aerosols sucked into the bypass duct are not destroyed there, both approaches yield the same time evolution of entrained aerosols (Fig. 9b), governed by the core air dilution factor $\mathcal{D}$ (Eq. (12)). Due to the continuous entrainment process, activated aerosols become abundant enough to significantly deplete the water vapor (difference between opaque and transparent curves in Fig. 9a) only at a stage when the relative humidity profiles of both approaches have already converged. This convergence occurs once the bypass and core flows have mixed and most of the kinetic energy has been converted into heat. Consequently, aerosols entrained after this point experience similar thermodynamic conditions in both modeling approaches, leading to similar ice crystal numbers.

## 5    Discussion

The following section discusses the presented findings in relation to existing literature. Based on this discussion, key implications are highlighted and possible directions for future research are proposed.

Schumann et al. (1997) pursued a similar idea to the approach outlined in Sec. 2.4. For kerosene combustion, they performed a large-eddy simulation that treated the core and bypass flow separately. They compared the temperature and partial pressure of water vapor of different plume parcels to the limiting mixing line of a core plume parcel that penetrates the bypass flow without significant interaction, thereby mixing directly with the cold ambient air. Their findings suggest that contrails could form under conditions where the classical Schmidt-Appleman theory (Schmidt, 1941; Appleman, 1953; Schumann, 1996) would not predict their formation.

Due to the higher energy-specific water vapor emission index in the hydrogen combustion case, the Schmidt-Appleman threshold temperature is not the decisive criterion for ice crystal formation, as this threshold temperature is typically higher than the homogeneous freezing temperature of supercooled droplets (Bier et al., 2024). Moreover, the penetrative mixing proposed by Schumann et al. (1997) is not necessarily needed for the onset of contrail formation for modern Turbofan engines: In the hydrogen combustion case, a plume parcel becomes already supersaturated with respect to water at an inverse dilution factor $\mathcal{D}^{-1} \sim 10$ (Fig. 2c). Clearly, the exact value depends on the specific engine exit and ambient conditions. This means that when the bypass ratio is higher than the inverse dilution factor value needed to cause supersaturation (which can be the case for modern high-bypass fans), the mixing of core and bypass air alone is sufficient to generate this supersaturation. As stated in the introductory part of Sec 2.4, this is also evident in the study by Richardson (2025) (his Fig. 4b). Nevertheless, in the Airbus Turbofan data, ambient air is already contained in the plume before the bypass air is thoroughly mixed into the core plume (Fig. 2c-d). This supports the findings by Schumann et al. (1997) and leads to even higher supersaturation and thus promotes the onset of ice crystal formation.

Previous studies (Schumann, 1996; Richardson, 2025) have derived an explicit formula for a mixing curve accounting for the kinetic energy in the plume. In their representations, the shape of the mixing curve is fully determined by the exit and ambient conditions and does not depend on the dilution speed. In their approaches, they assume that the exhaust is well-mixed at the engine exit, with the mean axial velocity that decreases linearly with the dilution. The kinetic energy is then computed based on





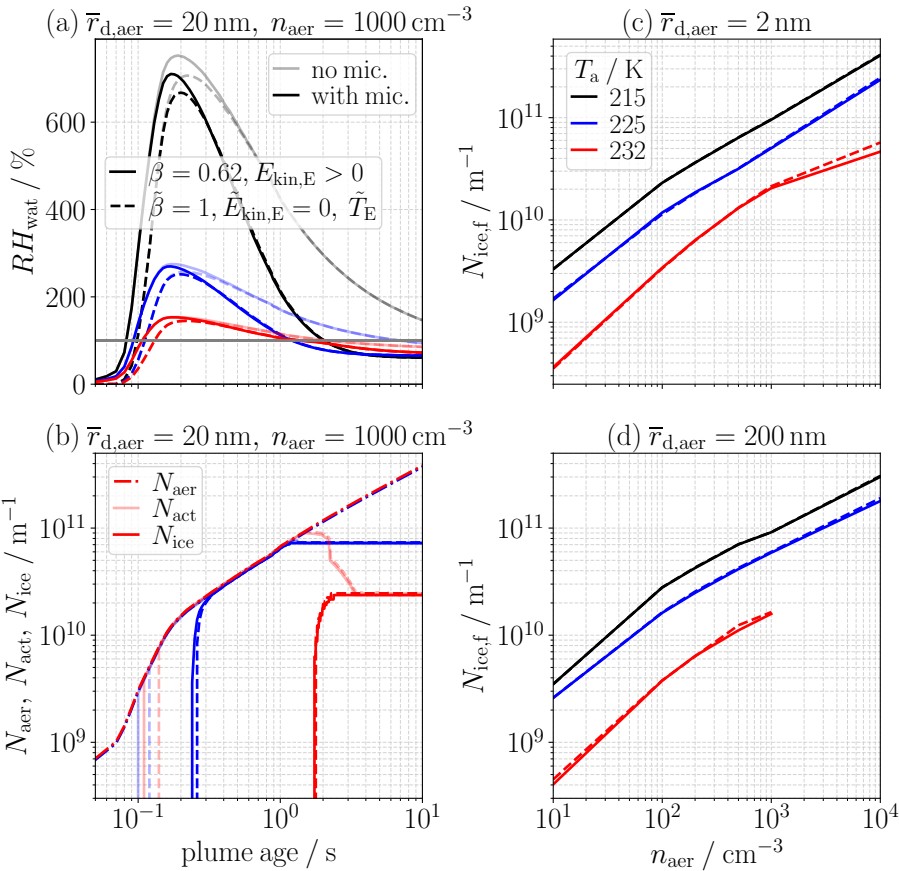

**Figure 9.** Results of box model simulations with the Airbus Turbofan dilution data. Compared are simulations with explicit thermodynamic treatment of bypass flow and kinetic energy (solid lines) to simulations that assume a stagnant exhaust plume with the emitted heat contained in the core flow at the engine exit (dashed lines). (a) Time evolution of relative humidity over water $RH_{\mathrm{wat}}$ without microphysical water vapor depletion (transparent lines) and with microphysics (opaque lines) for different ambient temperatures $T_{\mathrm{a}}$ (different colors). (b) Number of entrained aerosols $N_{\mathrm{aer}}$ (dashed-dotted lines), number of activated aerosols $N_{\mathrm{act}}$ (transparent lines) and number of ice crystals $N_{\mathrm{ice}}$ (opaque lines) as a function of plume age. For the sake of clarity, lines for $T_{\mathrm{a}} = 215\,\mathrm{K}$ are not shown. (c)-(d) Final number of ice crystals formed $N_{\mathrm{ice,f}}$ as a function of aerosol number concentration $n_{\mathrm{aer}}$ for two geometric mean radii $\overline{r}_{\mathrm{d,aer}}$. Results are shown for $N_{\mathrm{ice,f}} > 0$.

this mean velocity. Such an approach is appropriate when the radial profile of axial velocity is assumed to be uniform or has a shape that does not change over time (i.e., a self-similar profile (Pope, 2000)). However, the initial separation of total enthalpy into the core and bypass flow does not allow for such a treatment. The mixing of three air masses (core, bypass, ambient) with different properties results in a mixing curve that explicitly depends on the time evolution of a plume parcel's $\mathcal{D}$, $\gamma$ and $E_{\mathrm{kin}}$.

Clearly, the time-evolution of these quantities is influenced by the specific engine design, e.g., by the bypass ratio.





Additionally, Richardson (2025) examined cases with both constant and temperature-dependent specific heat capacities. The differences, however, are marginal compared to the changes we observe when the entrainment of bypass enthalpy is explicitly treated.

For entrained ambient aerosols, we have demonstrated that a simplified modeling approach is sufficient for turbofan engines (Sec. 4.5). This simplified approach assumes that the emitted combustion heat is fully contained as static enthalpy in the core flow at the engine exit. However, the simplified modeling approach might be insufficient for ice crystal formation on emitted particles in the case measurements at cruise altitude reveal a dominant contribution of lubrication oil particles (Ungeheuer et al., 2022; Ponsonby et al., 2024; Zink et al., 2025b) or clusters of nitric acid to the ice crystal formation process in the hydrogen combustion case. In contrast to ambient aerosols, these particles originating from emission products are not entrained over time into the plume but are present or have formed prior to contrail formation. As a consequence, a significant water vapor depletion by ice crystals formed on emitted particles typically occurs at an earlier stage (e.g., Fig. 3a in Zink et al. (2025b)) than in the case of ice crystal formation on ambient aerosols. At this stage, the bypass and core may not yet be thoroughly mixed, and the kinetic energy may not have been fully converted into heat. The higher possible supersaturations encountered compared to the simplified approach (Fig. 2f, Fig. 9a) might have an impact on the activation fractions. For conventional kerosene combustion, a recent study employing large-eddy simulations with a coupled microphysical model showed that the bypass flow influences the early contrail properties up to $\sim 0.6\,\mathrm{s}$ (Afkari et al., 2025). However, the final freezing fraction of soot particles is unaffected by the different bypass ratios they investigated.

Results from a large-eddy simulation showed that the mixing of different-aged plume parcels is a crucial process to be considered for ice crystal formation on emitted particles with high numbers (Lewellen, 2020). This process, however, is not represented in a simplified average box model approach. Therefore, if emitted particles dominate ice crystal formation, it may be beneficial to examine different parts of the core plume, specifically different trajectories that represent a certain fraction of the plume mass/volume. For a turbofan, each trajectory may follow an individual mixing curve depending on the individual time evolution of $\mathcal{D}$, $\gamma$ and $E_{\mathrm{kin}}$ (Eq. (28)). Such a trajectory ensemble, however, might only be beneficial when communication among the trajectories is included, accounting for the diffusive transport of water vapor and hydrometeors. Otherwise, the ice crystal number might be overestimated, as discussed in Bier et al. (2022).

For entrained ambient particles and kerosene combustion, Lewellen (2020) showed that neglecting the plume heterogeneity is not critical, and using an average box model approach yields ice crystal numbers comparable to those observed in large-eddy simulations. In addition, the LES results revealed a low sensitivity on different turbulence realizations. Our findings suggest that this low sensitivity on the exact mixing history is also valid for hydrogen combustion with contrail formation on ambient aerosols: Although the engine size scaling was initially applied to a single engine (Sec. 4.3), the associated dilution time scaling works also reasonable for dilution data from different sources (Sec. 4.4). These dilution data have different origins, utilizing various model approaches (RANS, LES, analytical) with different initial and boundary conditions, and differ in how the dilution factor is extracted (from temperature, core air fraction, or analytical). All these factors influencing the details of the time evolution of the dilution factor are of minor importance for the final number of ice crystals formed. Indeed, the sensitivity





to microphysical processes is much greater than the remaining differences observed among the various dilution data after the dilution time scaling has been applied (Fig. 3)

Moreover, the fact that the FLUDILES trajectory ensemble and average trajectory approaches yield similar ice crystal numbers, at least for temperatures where the homogeneous freezing timescale plays a secondary role (Fig. 7), additionally hints at the low sensitivity on the plume heterogeneity. The fact that the ice crystal numbers for these two approaches deviate for temperatures $T_\mathrm{a} \gtrsim 230\,\mathrm{K}$ hints that the plume heterogeneity may be important when the homogeneous freezing significantly influences the number of ice crystals. The trajectory ensemble without inter-trajectory communication, however, may not capture the entire physics. Therefore, forward approaches should be either to incorporate the communication into the FLUDILES trajectory data set or to investigate the homogeneous freezing dependency with RadMod (Lottermoser and Unterstrasser, 2024) coupled to LCM.

## 6 Conclusions

We extended the LCM box model by integrating several aspects related to the aircraft engine. This extended version was then used to simulate contrail formation on entrained ambient aerosols for hydrogen combustion. The results provide a basis for parameterizing the number of ice crystals formed $N_\mathrm{ice,f}$. Such a parametrization can be integrated into a general circulation model or any other large-scale contrail model to assess the radiative impacts of contrails originating from a fleet of aircraft with hydrogen combustion.

The key findings of our study are:

– Changes in engine exit conditions and dilution speed due to changing ambient temperature have a negligible effect on the formation of ice crystals. These influences can be safely disregarded in a parametrization.

– A change in overall propulsion efficiency has approximately the same effect on the thermodynamic plume evolution as flying at a different pressure level while keeping the overall propulsion efficiency fixed. Consequently, an accurate representation of the ambient pressure dependency inherently accounts for the dependence on overall propulsion efficiency. Therefore, we can express the sensitivity of $N_\mathrm{ice,f}$ to the overall propulsion efficiency through its sensitivity to the ambient pressure.

– The ice crystal numbers resulting from different-sized engines and different exit jet speeds can be scaled to each other. Therefore, it is sufficient to accurately represent $N_\mathrm{ice,f}$ for one engine size. The ice crystal number for another-sized engine can then be obtained by appropriate scaling.

– It is sufficient to assume that the whole emitted combustion heat is contained as static enthalpy in the core flow at engine exit. This simplified approach yields similar ice crystal numbers as the approach with explicit treatment of kinetic energy dissipation and entrainment of enthalpy initially contained in the bypass flow of a turbofan engine.





We note that these conclusions are valid for contrail formation on entrained ambient aerosols and not all conclusions are necessarily also true for kerosene combustion or scenarios where ice crystals form predominantly on plume particles (i.e., those particles originating from emitted species).

If other ice crystal formation pathways in the hydrogen combustion case (e.g. on lubrication oil particles) will show to dominate over the formation on ambient aerosol, simulations incorporating these potential nuclei would need to be conducted

to reassess the aforementioned conclusions.

## Appendix A

The total temperature (or stagnation temperature) $T_{\mathrm{tot}}$ and the static temperature $T_{\mathrm{static}}$ are related via (Walsh and Fletcher, 2004)

$$T_{\mathrm{tot}} = T_{\mathrm{static}} \left( 1 + \frac{\kappa - 1}{2} Ma^2 \right) . \tag{A1}$$

Neglecting the slight temperature dependence of the adiabatic index $\kappa$ and preserving the Mach number $Ma$, scaling $T_{\mathrm{tot}}$ by a factor $c$ means that also the static temperature $T_{\mathrm{static}}$ scales with this factor $c$.

The total pressure $p_{\mathrm{tot}}$ and static pressure $p_{\mathrm{static}}$ for a compressible isentropic flow are related by (Walsh and Fletcher, 2004)

$$p_{\mathrm{tot}} = p_{\mathrm{static}} \left( 1 + \frac{\kappa - 1}{2} Ma^2 \right)^{\frac{\kappa}{\kappa - 1}} \tag{A2}$$

As for the temperatures, Eq. (A2) implies that total and static pressures scale the same way (assuming constant $\kappa$).

## 685    Appendix B

Lewellen (2020) introduced two timescales: The first is the time interval during which a plume parcel can potentially activate aerosols, which he called $\tau_{\mathrm{mix}}$ and has basically the same meaning as $\delta t_{\mathrm{supersat}}$ we used in Sec. 4.3. For this timescale it holds $\tau_{\mathrm{mix}} \propto s_{\mathrm{dil}}$. The second is the time interval needed for the ice crystals to grow large enough to substantially consume the water vapor, for which Lewellen (2020) used the phase relaxation time scale $\tau_{\mathrm{fc}}$ after Khvorostyanov and Sassen (1998). It

holds $\tau_{\mathrm{fc}} \propto (\bar{r}_{\mathrm{ice}} n_{\mathrm{aer}})^{-1}$ and since the mean radius of the ice crystals scales as $\bar{r}_{\mathrm{ice}} \propto n_{\mathrm{aer}}^{-1/3}$, it follows $\tau_{\mathrm{fc}} \propto n_{\mathrm{aer}}^{-2/3}$. Lewellen (2020) showed that when plotting the apparent ice particle emission index normalized by the aerosol number concentration $EI_{\mathrm{ice}}/n_{\mathrm{aer}}$ against any function of the ratio of those two timescales, e.g., $(\tau_{\mathrm{mix}}/\tau_{\mathrm{fc}})^{3/2} \propto s_{\mathrm{dil}}^{3/2} n_{\mathrm{aer}}$, the curves for different engine sizes/dilution speeds collapse. Mathematically this can be expressed as

$$\frac{EI_{\mathrm{ice}}}{n_{\mathrm{aer}}} \sim g\big(s_{\mathrm{dil}}^{3/2} n_{\mathrm{aer}}\big) \ , \tag{B1}$$

where $g$ is a function that depends on ambient conditions and aerosol properties but not on the engine size/dilution speed. Since $N_{\mathrm{ice,f}} \propto s_{\mathrm{E}}^2 EI_{\mathrm{ice}}$, it exists a function $\tilde{g}$ fulfilling

$$\frac{N_{\mathrm{ice,f}}}{s_{\mathrm{E}}^2 n_{\mathrm{aer}}} \sim \tilde{g}\big(s_{\mathrm{dil}}^{3/2} n_{\mathrm{aer}}\big) \ . \tag{B2}$$



Multiplication of Eq. (B2) with $s_{\mathrm{dil}}^{3/2} n_{\mathrm{aer}}$ and introducing the function $f(x) = x\tilde{g}(x)$ leads to Eq. (37) with $a = -3/2$.

*Data availability.* The presented simulation results are available from the corresponding author upon request (josef.zink@dlr.de)

*Author contributions.* **Josef Zink**: Conceptualization, Data curation, Formal analysis, Investigation, Methodology, Software, Validation, Visualization, Writing – original draft, Writing – review & editing, **Simon Unterstrasser**: Conceptualization, Methodology, Funding acquisition, Software, Supervision, Project Administration, Writing – review & editing

*Competing interests.* The contact author has declared that none of the authors has any competing interests.

*Acknowledgements.* This work has been funded by Airbus and by the DLR internal project "H2CONTRAIL". We thank X. Vancassel
(ONERA), D. Lewellen and Airbus for providing CFD data on plume dilution. We thank U. Schumann for an internal review of the paper draft. Furthermore, we thank C. Renard and J. Purseed for their comments. The language was polished with the help of ChatGPT.



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
