# Peer review of "Contrail formation for aircraft with hydrogen combustion - Part 2: Engine-related aspects"

_EGUsphere, 2025_

## Author Comment (AC1)

Dear editor and reviewers,

We thank the editor for handling our manuscript, and we are grateful to both reviewers for their careful reading of the manuscript and for the thoughtful comments. We have addressed all comments and revised the manuscript accordingly.
In this response document, reviewer comments are reproduced in black. Our replies are provided in green, and references to the manuscript are indicated by *'green italicized text in quotation marks'*.

**Reviewer 1 (RC1)**

This is a detailed and comprehensive analysis of how the engine parameters of bypass, propulsive efficiency, and engine size end up affecting how contrails form. This is a significant and important advance over related prior work in this area and offers useful insights for how contrails will form when $H_2$ is used as the fuel. Many of the results also apply to kerosene-fueled engines, but the authors are careful to highlight in which ways the conclusions may not hold for kerosene combustion.

The conclusions of the analysis are important in that, in many cases, some of the details of the complex exhaust mixing are not important and simplifying model assumptions can be made. This will be a boon for further analysis. The conclusion that overall propulsive efficiency can be understood by considering flight at a different ambient pressure is not intuitively obvious (at least to me) but is also a useful conclusion for model simplification. Again, many of these conclusions apply regardless of the kerosene/$H_2$ fuel difference, and the authors point out where the conclusions may not apply to kerosene combustion.

I have no major criticisms of the manuscript which is well-structured and clearly written throughout. I will list a few minor comments that the authors may decide if they would care to address.

(and several of these perhaps relate more to ChatGPT than the authors!)

1. I thought that the spelling "parameterizations" (on pages 1 and 2) was the more accepted/common spelling for that term, even though "parametrizations" seems to an acceptable alternative.
   Many thanks for hinting at this inconsistency. We now consistently use the more common spelling *'parameterization'* throughout the manuscript.

2. Line 431: "whether these neglects . . ." is an unusual grammatical construct. It is probably not confusing, but might read better as "whether neglecting these changes . . ."
   Thanks. We changed it accordingly.

3. Line 164, equation (9): Given the attention to detail and rigorous nature of the analysis, this approximation jumped out at me. While the air-to-fuel ratio is a large number even for kerosene combustion, presumably (for the conditions chosen for the $H_2$ combustion case) this is an even larger number here. Perhaps a word or two and/or a rough air-to-fuel number for this $H_2$ combustion case might be useful to justify this approximation (since it probably would not be much computational burden to carry the full expression forward). I am sure that the approximation does not impact any conclusions at all, but the approximation is just summarily included without any justification as things stand.
   You are right. We added the following for clarification:
   *'…where the fuel mass is neglected in the approximation due to the typically large air-to-fuel ratios. For kerosene combustion, air-to-fuel ratios are on the order of 50–70 (Kärcher et al., 2015). These ratios may be even higher for hydrogen combustion, because the mass-specific*

*heat of combustion is larger by a factor of 2.79 (Bier et al., 2024), implying that less fuel mass is required to produce the same amount of energy.'*

4. Line 541, "does not contain an explicit dependence on engine size" would be preferable?
   We changed *'dependency'* to *'dependence'*.

5. Lines 612 on, and Lines 673 on: When oil is discussed, it seems to implicitly assume that the oil emissions are included in the core flow. That is true for some engines, but certainly not all. Since the authors are introducing the subject of oil and their extensive analysis differentiates exhaust emissions and entrained nuclei, I think a more nuanced discussion should be offered. While GE/CFM engines release oil breather flow into the core flow, many other engines release oil breather flow into the bypass or external to the nacelle. Especially for the nacelle release location, the oil would be entrained with ambient air and so would be more like ambient nuclei entrainment (although all at a very early location for oil, as opposed the continual downstream entrainment of ambient nuclei). The bypass oil breather case would be somewhere in between.

- Since this analysis also pays strict attention to particle size distributions, it is also worth noting that oil breather flow being released in the hot core exhaust vaporizes the oil that then recondenses in small nucleation mode particles. Conversely, release that is external to the nacelle results in oil particles with a few hundred nm sized oil droplets.

- The authors can decide how much additional detail they wish to say about oil, but the current language suggesting that oil should be considered an exhaust emission is misleading and incorrect for some engines.

  You are right, the text as written may give the impression that lubrication oil poses the same concern for all engines. In fact, a large impact of lubrication oil is only expected when venting occurs close to the core flow, where the oil may evaporate and subsequently form large numbers of small particles. This renucleation process may be completed well before contrail formation starts, so for these cases the oil particles can be considered as emitted particles. By contrast, oil particles with sizes of a few hundred nanometers that are released external to the nacelle and entrained at a stage when the plume has already cooled down are expected to be very low in number.

  To clarify that our text mainly refers to engines with venting close to the core, we changed the sentences in the discussion part to:
  *'In the hydrogen combustion case, possible sources of such particles are nitric acid formed through oxidation of NOx emissions or lubrication oil (Ungeheuer et al., 2022; Ponsonby et al., 2024; Zink et al., 2025b). For engines where the lubrication system vents near the core flow, a large number of small particles can form when oil evaporates in the hot sections and subsequently re-nucleates in the cooling plume (Ungeheuer et al., 2022).'*

  In the conclusion we extended the sentence to:
  *"In case other ice crystal formation pathways in the hydrogen combustion case turn out to dominate over the formation on ambient aerosol (e.g., if lubrication oil enters the hot exhaust regions, where it may evaporate and subsequently form a distribution of small particles with high numbers), simulations incorporating these potential nuclei would need to be conducted to reassess the aforementioned conclusions"*
  In Part 3, which we will submit soon, we provide a longer discussion of this topic. There we will specifically address your points regarding the release location and the size distribution.

**Reviewer 2 (RC2)**
This work describes a deep evaluation of some ways in which engine design might affect the long-term behavior of a contrail, focusing on the intermediate quantity of the number of ice crystals formed. The work is timely, important, and – since it primarily concerns hydrogen engines – of great immediate value to both the contrail science and aeronautical engineering communities.

The central question posed – whether engine design can affect contrail impacts – is both interesting and important. The methods used are appropriate, and the conclusions are well supported by the data produced. This work constitutes a useful methodological advance while also bringing a greater level of technical depth to bear on the question of how contrails from LH2-fuelled aircraft might behave.

I have no methodological concerns, and relatively few comments on the manuscript; those are listed below.

My biggest question relates to the assumed properties of the ambient aerosols. Is it accurate to say that the ambient aerosol concentration is more strictly the concentration of ambient aerosols which it can be assumed would be effective nuclei for contrail ice? Given the amount of recent research dedicated to the question of efficacy of soot in this regard, it would be useful if the authors could comment on what they are assuming about the ambient particles and how the assumed ambient aerosol number would relate to typical measurements taken in the upper troposphere.

You are probably referring to the studies by Mahrt et al (2020) and Kärcher et al (2021) who discuss the ice-forming capabilities of pre-processed aircraft soot particles and its relevance for natural cirrus formation.

Clearly, aviation soot particles will be present in the environment as aircraft that use fossil fuels are certainly used in parallel to hydrogen-powered aircraft. The entrainment of these soot particles into the plume is then the same as for all other ambient particles. Entrained ambient aerosol particles are assumed to be activated into liquid droplets with subsequent homogenous freezing into ice crystals. This formation pathway is justified by the fact that cloud nuclei (CN) are much more frequent in the upper troposphere than ice nuclei (IN) (e.g., Rogers et al., 1998; Beer et al., 2022).
Clearly, the papers listed above discuss the relevance of aviation soot for natural cirrus formation, as it may increase largely the number of available INs. For contrail formation, however, the number of the much more abundant CNs is relevant.

In Part 1, which we submitted in parallel, the focus is strongly on the ambient aerosol properties. There a dedicated section about properties of ambient aerosol in the upper troposphere are given based on measurement results. These measurements show a large variability in number concentration and other properties of ambient aerosols (size, solubility). Various scenarios are investigated in Part 1, including also those ambient aerosols with properties representative of soot particles (having low solubility). The focus of the current study (Part 2) is on engine-related aspects. But we still varied the number concentration of ambient aerosol particles over a wide range of potential scenarios (Fig. 6, 7, 8, 9) including rather low and high values.

We added following sentences to the end of Sec. 3 for clarification and with the hint to Part 1:
*'The prescribed baseline properties of ambient aerosol particles correspond to highly-soluble Aitken-mode particles. Ambient aerosol particle sizes, solubility, and number concentrations show large variability in the atmosphere (Petzold et al., 2002; Minikin et al., 2003; Hermann et al., 2003; Borrmann et al., 2010; Kaiser et al., 2019; Brock et al., 2021). In Part 1 (Zink et al., 2025a), we systematically examined how these properties influence droplet activation and the subsequent*

*homogeneous freezing of droplets into ice crystals. Although the primary focus of the current study is on engine-related parameters, we nonetheless examine a variety of ambient aerosol scenarios, covering a wide range of number concentrations (Sect. 4).'*

I believe the statement on line 631 is an accidental double negative: "Lewellen (2020) showed that neglecting the plume heterogeneity is not critical". Should "neglecting" here instead be (e.g.) "accounting for"?

The wording is intentional: Lewellen (2020) showed that, for entrained ambient aerosols, neglecting plume heterogeneity does not critically affect the results, which justifies the use of a simplified 0D box model that does not resolve plume heterogeneity.

There are some minor grammatical errors throughout (e.g. "whether these neglects", line 431; "low sensitivity on" (rather than to), line 633; "will show to dominate", line 674). I would recommend that the authors perform an additional sweep to remove these.

Thanks. We changed it accordingly.

The abstract currently provides little actionable information, and I found myself unable to gather much insight from it. Given that the authors have compiled a compelling set of findings in the conclusions, I would recommend at least summarizing some of these in the abstract.

You are right. We added following sentences to the abstract.

'*We find that the impact of a change in overall efficiency can be mimicked by adjusting the ambient pressure. Moreover, results from scenarios with different engine sizes or jet speeds can be scaled onto each other. Furthermore, for contrail formation on entrained ambient aerosols, a simplified modeling approach is sufficient, assuming that all emitted combustion heat is contained as static enthalpy in the core flow at engine exit.*'

My last comment is a plea rather than a concern, but I would nonetheless be very happy to see addressed. As has become unfortunately common in the contrail literature, the authors refer to the "overall propulsion efficiency" (see e.g. Equation 2). This is confusing terminology, as standard textbooks on engine design (see e.g. Hill and Peterson, or Cumpsty) define both an overall engine efficiency (ratio of thrust power output to heat input – what is intended here) and an engine propulsion efficiency (ratio of thrust power output to the rate of production of propellant kinetic energy – decidedly not what is intended here). The distinction is important, because the propulsive efficiency is (by definition) always greater than or equal to the overall efficiency and, if used by accident by a reader of this manuscript, would change the meaning of the equations. Given that I have had to correct many confused manuscripts, students, and colleagues that have accidentally applied the wrong definition as a consequence of the term "overall propulsion efficiency" being used in prior contrail literature, I would be very grateful if the authors would avoid propagating this issue. This can be fixed by explicitly stating that they are using the "engine overall efficiency" and avoiding use of the term "propulsion efficiency" except when they actually mean the propulsion (or propulsive) efficiency.

We thank the reviewer for this thoughtful comment. As stated in the Introduction, we defined the term *overall propulsion efficiency* as the product of *thermal efficiency* and *propulsive efficiency*, and we had assumed this terminology to be standard within the contrail literature, which is the primary context of our work. We recognize, however, that the use of the word *propulsion* may cause confusion, particularly when viewed from the perspective of classical engine-design terminology.

At the same time, we note that the alternative term *engine overall efficiency* is, in our view, also potentially misleading, as it suggests that the quantity depends solely on the engine. As emphasized by Schumann (2000):

"Since the overall efficiency depends on speed V, and the thrust F balances the aircraft drag, it is actually not a parameter of the engine alone but characterises the engine/aircraft combination and its state of operation."

To address the reviewer's concern while avoiding this ambiguity, we propose to use the term *overall efficiency of propulsion.* By placing *propulsion* after *efficiency*, the term is less likely to be confused with *propulsive efficiency*, while preserving the established meaning used in the contrail literature. In places where the meaning is unambiguous, we will use the shorter term *overall efficiency*. We believe this represents a pragmatic compromise that improves clarity without confusing the contrail community.

**References:**

Rogers, D. C., DeMott, P. J., Kreidenweis, S. M., and Chen, Y.: Measurements of ice nucleating aerosols during SUCCESS, Geophys. Res. Lett., 25, 1383–1386, https://doi.org/10.1029/97gl03478, 1998.

Beer, C. G., Hendricks, J., and Righi, M.: A global climatology of ice-nucleating particles under cirrus conditions derived from model simulations with MADE3 in EMAC, Atmos. Chem. Phys., 22, 15 887–15 907, https://doi.org/10.5194/acp-22-15887-2022, 2022.

Schumann, U.: Influence of propulsion efficiency on contrail formation, Aerosp. Sci. Technol., 4, 391–402, 2000.

Mahrt, F., Kilchhofer, K. , Marcolli, C., Grönquist, P. , David, R. O., Rösch, M. , Lohmann, U. . Kanji, Z. A., The Impact of Cloud Processing on the Ice Nucleation Abilities of Soot Particles at Cirrus Temperatures, 2020

Kärcher, B. Mahrt, F.,  Marcolli, C., Process-oriented analysis of aircraft soot-cirrus interactions constrains the climate impact of aviation, 2021, *Communications Earth & Environment* , Vol. 2, No. 1, p. 113